# Peptidoglycan precursor synthesis along the sidewall of pole-growing mycobacteria

**Alam García-Heredia[1†], Amol Arunrao Pohane[2†], Emily S Melzer[2], Caleb R Carr[2], Taylor J Fiolek[3], Sarah R Rundell[3], Hoong Chuin Lim[4], Jeffrey C Wagner[5], Yasu S Morita[1,2], Benjamin M Swarts[3], M Sloan Siegrist[1,2]***

[1]Molecular and Cellular Biology Graduate Program, University of Massachusetts, Amherst, United States; [2]Department of Microbiology, University of Massachusetts, Amherst, United States; [3]Department of Chemistry and Biochemistry, Central Michigan University, Mount Pleasant, United States; [4]Department of Microbiology and Immunobiology, Harvard Medical School, Boston, United States; [5]Department of Immunology and Infectious Disease, Harvard T.H. Chan School of Public Health, Boston, United States

**\*For correspondence:**
siegrist@umass.edu

[†]These authors contributed equally to this work

**Competing interests:** The authors declare that no competing interests exist.

**Abstract** Rod-shaped mycobacteria expand from their poles, yet D-amino acid probes label cell wall peptidoglycan in this genus at both the poles and sidewall. We sought to clarify the metabolic fates of these probes. Monopeptide incorporation was decreased by antibiotics that block peptidoglycan synthesis or L,D-transpeptidation and in an L,D-transpeptidase mutant. Dipeptides complemented defects in D-alanine synthesis or ligation and were present in lipid-linked peptidoglycan precursors. Characterizing probe uptake pathways allowed us to localize peptidoglycan metabolism with precision: monopeptide-marked L,D-transpeptidase remodeling and dipeptide-marked synthesis were coincident with mycomembrane metabolism at the poles, septum and sidewall. Fluorescent pencillin-marked D,D-transpeptidation around the cell perimeter further suggested that the mycobacterial sidewall is a site of cell wall assembly. While polar peptidoglycan synthesis was associated with cell elongation, sidewall synthesis responded to cell wall damage. Peptidoglycan editing along the sidewall may support cell wall robustness in pole-growing mycobacteria.
DOI: https://doi.org/10.7554/eLife.37243.001

## Introduction

Model, rod-shaped organisms such as *Escherichia coli* and *Bacillus subtilis* elongate across a broad swath of the cell (*de Pedro et al., 1997*; *Daniel and Errington, 2003*). Mycobacterial cells, by contrast, extend from narrower polar regions (*Aldridge et al., 2012*; *Santi et al., 2013*; *Meniche et al., 2014*; *Thanky et al., 2007*; *Kieser and Rubin, 2014*; *Singh et al., 2013*; *Joyce et al., 2012*). Circumscription of growth to discrete zones poses spatial challenges to the bacterial cell. For example, if polar growth and division are the only sites of cell wall synthesis in mycobacteria, the entire lateral surface of the cell must be inert (*Aldridge et al., 2012*; *Brown et al., 2012*; *Kuru et al., 2012*; *Zupan et al., 2013*). Such an expanse of non-renewable surface could leave the cell vulnerable to environmental or immune insults.

Because cell wall peptidoglycan synthesis is critical for bacterial replication, it is often used to localize the sites of growth and division. Intriguingly, D-amino acid probes, which in other species have been shown to incorporate into peptidoglycan (*de Pedro et al., 1997*; *Kuru et al., 2012*; *Siegrist et al., 2013*), label both the poles and sidewall of mycobacteria (*Meniche et al., 2014*;

Siegrist et al., 2013; Boutte et al., 2016; Botella et al., 2017; Schubert et al., 2017; Rodriguez-Rivera et al., 2018). The localization of these molecules is supported by the detection of peptidoglycan synthetic enzymes at the mycobacterial cell tips and periphery (Meniche et al., 2014; Joyce et al., 2012; Hett et al., 2010; Kieser et al., 2015a; Plocinski et al., 2011). However, both intracellular and extracellular incorporation pathways have been characterized or hypothesized for D-amino acid probes, complicating the interpretation of labeling patterns (Siegrist et al., 2015). Intracellular uptake implies that the probe enters the biosynthetic pathway at an early stage, and therefore marks nascent cell wall. Extracellular incorporation, on the other hand, suggests that the probe enters the pathway at a later stage and/or is part of enzymatic remodeling of the macromolecule in question. The extent to which peptidoglycan synthesis and remodeling are linked is not clear (Brown et al., 2012; de Pedro and Cava, 2015; Glauner and Höltje, 1990) and may vary with species and external milieu. In *Mycobacterium tuberculosis*, for example, there is indirect but abundant data that suggest that there is a substantial cell envelope remodeling during infection when growth and peptidoglycan synthesis are presumed to be slow or nonexistent (Kieser and Rubin, 2014).

An intracellular metabolic tagging method for the cell wall would be an ideal tool for determining whether tip-extending mycobacteria can synthesize peptidoglycan along their lateral surfaces. At least two pieces of evidence suggest that D-alanine-D-alanine dipeptide probes are incorporated into peptidoglycan via the cytoplasmic MurF ligase (Liechti et al., 2014; Sarkar et al., 2016). First, derivatives of D-alanine-D-alanine rescue the growth of *Chlamydia trachomatis* treated with D-cycloserine, an antibiotic that inhibits peptidoglycan synthesis by inhibiting the production and self-ligation of D-alanine in the cytoplasm (Liechti et al., 2014). Second, *B. subtilis* cells stripped of mature peptidoglycan by lysozyme treatment retain a small amount of dipeptide-derived fluorescence (Sarkar et al., 2016). While these data are suggestive, formal demonstration of intracellular incorporation requires direct evidence that the probe labels peptidoglycan precursors. More broadly, better characterization of the metabolic fate of probes would increase the precision of conclusions that can be drawn from labeling experiments (Boyce et al., 2011; Qin et al., 2017).

Here, we sought to determine how D-amino acid probes incorporate into the mycobacterial cell wall. Monopeptide D-amino acid probes chiefly reported peptidoglycan remodeling by L,D-transpeptidases while dipeptides marked lipid-linked peptidoglycan precursors. All the probes tested labeled the poles and sidewall of mycobacteria, indicating that cell wall metabolism in these regions comprises both synthetic and remodeling reactions. While peptidoglycan assembly along the mycobacterial periphery did not support obvious surface expansion, it was greatly enhanced by cell wall damage. Such activity may allow editing of a complex, essential structure at timescales faster than those permitted by polar growth.

## Results

### Metabolic labeling of mycobacterial envelope comprises asymmetric polar gradients

Mycobacteria have been shown to expand from their poles (Aldridge et al., 2012; Santi et al., 2013; Meniche et al., 2014; Thanky et al., 2007; Kieser and Rubin, 2014; Singh et al., 2013; Joyce et al., 2012) but published micrographs suggest that D-amino acid probes may label both the poles and sidewall of these organisms (Meniche et al., 2014; Siegrist et al., 2013; Boutte et al., 2016; Botella et al., 2017; Schubert et al., 2017; Rodriguez-Rivera et al., 2018). Metabolic labeling can be achieved by a one-step process, in which the fluorophore is directly appended to the probe, or a two-step process in which a small chemical tag on the D-amino acid is detected by subsequent reaction with a fluorescent reactive partner ([Siegrist et al., 2015], Figure 1A). We first reexamined the localization of various D-amino acid probes reported in the literature, including RADA (Kuru et al., 2012), which is directly conjugated to 5-carboxytetramethylrhodamine, and the two-step alkyne-D-alanine [alkDA or EDA, (Kuru et al., 2012; Siegrist et al., 2013)], and alkyne-D-alanine-D-alanine (alkDADA or EDA-DA, (Liechti et al., 2014); we use the metabolic labeling nomenclature originally adopted in [Mahal et al., 1997]) which we detected by copper-catalyzed azide-alkyne cycloaddition (CuAAC) after fixation (Figure 1B and C). After *M. smegmatis* cells were incubated for ~10% generation in probe, high-resolution and quantitative microscopy revealed that they had asymmetric, bidirectional gradients of fluorescence that emanated from the poles and continued

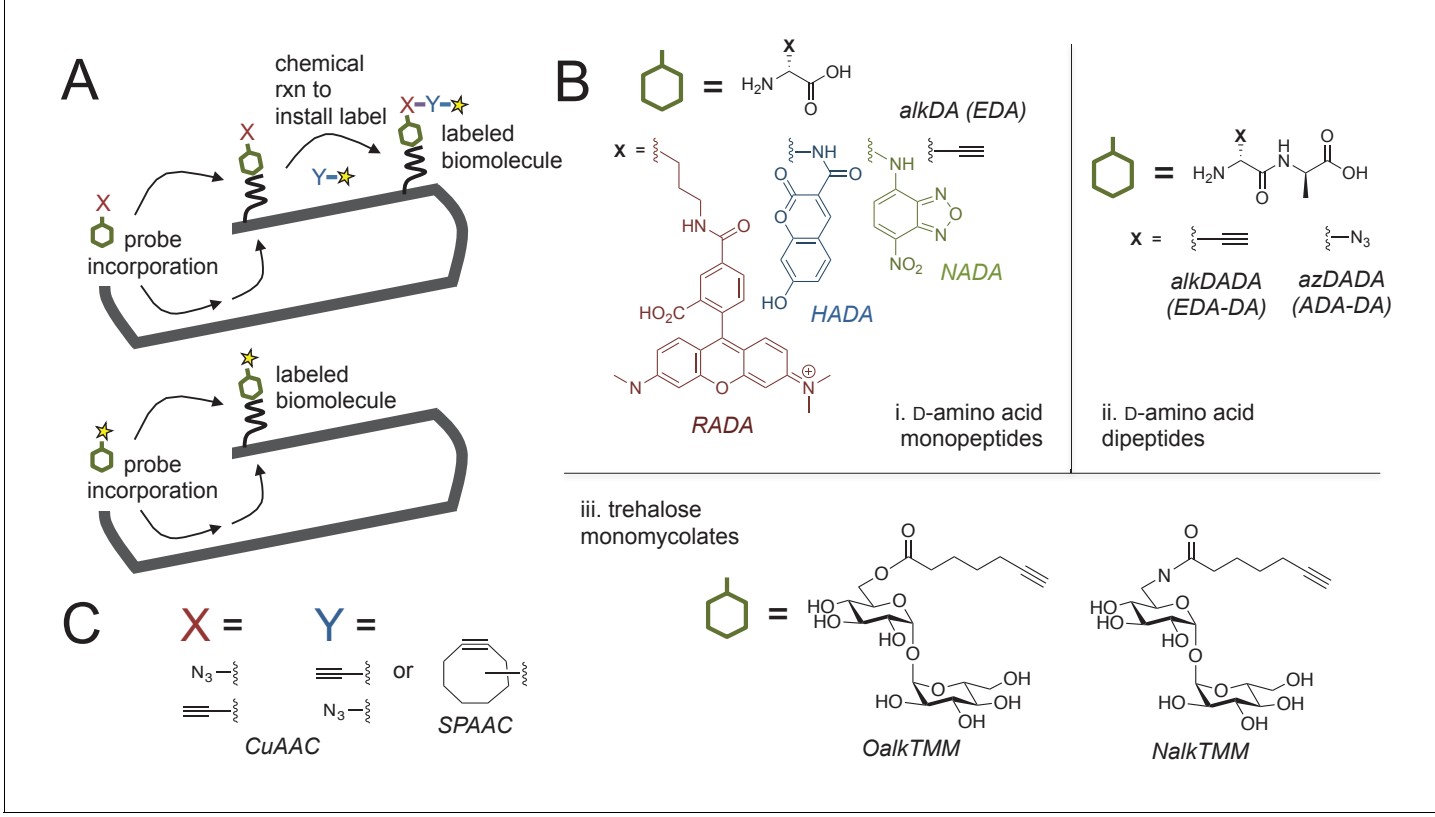

**Figure 1.** Cell envelope metabolic labeling in mycobacteria. (**A**) Schematic of one- and two-step metabolic labeling. Top, a cell envelope precursor or 'probe' bearing a reactive group is incorporated into the envelope by the endogenous enzymatic machinery of the cell. The presence of the probe is then revealed by a chemical reaction with a label that bears a complementary reactive group. Bottom, in some cases the probe can be pre-labeled, bypassing the chemical ligation step and embedding the detection moiety directly into the macromolecule. Yellow star, fluorophore. See (*Siegrist et al., 2015*) for more details. (**B**) Probes used in this work to mark the mycobacterial envelope. See text for details. Colored and black chemical structures denote probes used in one- and two-step labeling, respectively. C, X and Y reactive partners used in this work for two-step labeling as shown in A. CuAAC, copper-catalyzed azide-alkyne cycloaddition; SPAAC, strain-promoted azide-alkyne cycloaddition.
DOI: https://doi.org/10.7554/eLife.37243.002

along the sidewall (*Figure 2A*). Polar gradients of dipeptide labeling were also apparent in live cells when we detected azido-D-alanine-D-alanine (azDADA or ADA-DA, [*Liechti et al., 2014*]) incorporation by either CuAAC (using low copper, bio-friendly reaction conditions ([*Yang et al., 2014*], *Figure 2—figure supplement 1A*) or by copper-free, strain-promoted azide-alkyne cycloaddition (SPAAC, *Figure 2—figure supplement 1B*).

The mycobacterial cell envelope is comprised of covalently bound peptidoglycan, arabinogalactan and mycolic acids, as well as intercalated glycolipids and a thick capsule ([*Puffal et al., 2018*], *Figure 1B*). Assembly of the envelope layers has long been presumed to be spatially coincident. This is largely based on biochemical data suggesting that ligation of arabinogalactan to peptidoglycan occurs concurrently with crosslinking of the latter by transpeptidases (*Hancock et al., 2002*). We and others have found that cytoplasmic enzymes that mediate arabinogalactan and mycomembrane synthesis are enriched at the poles but also present along the periphery of the cell (*Meniche et al., 2014*; *Hayashi et al., 2016*; *Carel et al., 2014*), as is metabolic labeling by OalkTMM and NalkTMM ([*Backus et al., 2011*; *Swarts et al., 2012*; *Foley et al., 2016*], *Figure 2A*). OalkTMM and NalkTMM are trehalose monomycolate derivatives that predominantly mark covalent mycolates and trehalose dimycolate, respectively, in the mycomembrane (*Figure 1B*, [*Foley et al., 2016*]). The azido and alkynyl groups on the different probes are not orthogonal to each other (*Figure 1C*) so we opted to compare peptidoglycan and mycomembrane labeling patterns by using RADA as a fiducial marker. The cell pole with brighter RADA fluorescence also had more alkDADA or OalkTMM labeling (*Figure 2B*), suggesting that the polar orientation of peptidoglycan and mycomembrane metabolism

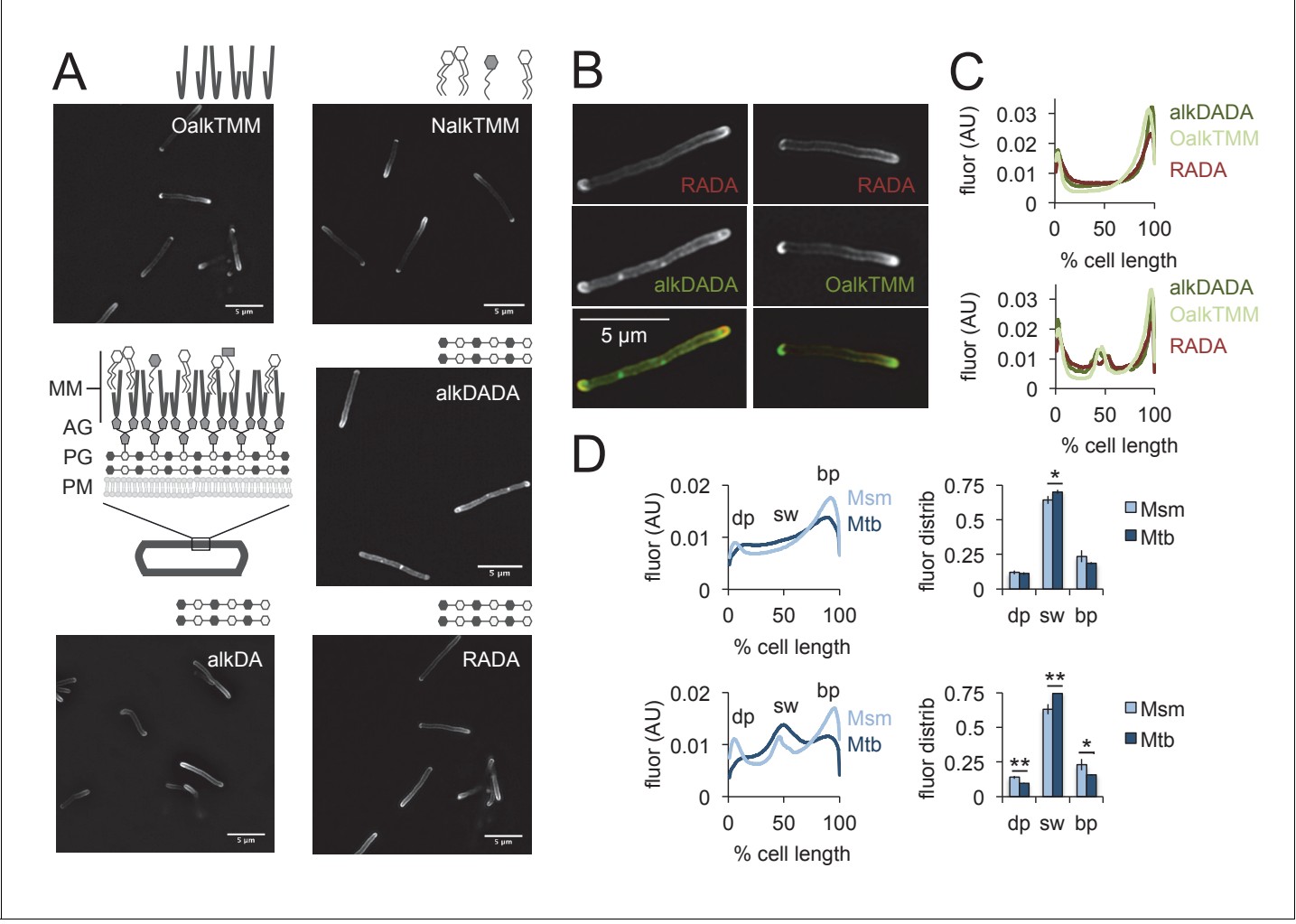

**Figure 2.** Asymmetric polar gradients of cell envelope metabolic labeling in mycobacteria. (A) *M. smegmatis* was incubated for 15 min (~10% generation) in the indicated probe, then washed and fixed. Alkynyl probes were detected by CuAAC with azido-CR110 and cells were imaged by structured illumination microscopy. MM, mycomembrane; AG, arabinogalactan; PG, peptidoglycan; PM, plasma membrane. (B) *M. smegmatis* dual labeled with RADA and alkDADA, left, or RADA and OalkTMM, right, and imaged by conventional fluorescence microscopy. (C) *M. smegmatis* was labeled as in B and cellular fluorescence was quantitated for cells without (top; 77 < n < 85) or with (bottom; 9 < n < 51) visible septa for RADA, OalkTMM and alkDADA. Signal was normalized to cell length and to total fluorescence intensity. Cells were oriented such that the brighter pole is on the right hand side of the graph. (D) *M. smegmatis* (Msm; light blue) or *M. tuberculosis* (Mtb; dark blue) was labeled with HADA for 15 min or 2 hr (~10% generation), respectively, then washed and fixed. Fluorescence was quantitated as in C for cells without (top; 118 < n < 332) and with (bottom; 55 < n < 85) visible septa. We defined the dim pole (dp) as the sum of the fluorescence intensity over the first 15% of the cell; the sidewall (sw) as the sum of the middle 70%; and the bright pole (bp) as the sum over the final 15% of the cell. Fluor distrib, average fluorescence distribution. AU, arbitrary units. Error bars, ±standard deviation. Statistical significance between *M. smegmatis* (five biological replicates) and *M. tuberculosis* (three biological replicates) was assessed for the dim pole, sidewall and bright pole by two-tailed Student's t test. *p<0.05; **p<0.005.

DOI: https://doi.org/10.7554/eLife.37243.003

The following source data and figure supplements are available for figure 2:

**Source data 1.** Conventional fluorescence and structured illumination microscopy images of mycobacteria labeled with peptidoglycan and trehalose monomyocolate probes.

DOI: https://doi.org/10.7554/eLife.37243.008

**Figure supplement 1.** Metabolic labeling by azDADA comprises polar gradients in live *M. smegmatis*.

DOI: https://doi.org/10.7554/eLife.37243.004

**Figure supplement 1—source data 1.** Conventional fluorescence microscopy images of AzDADA-labeled *M. smegmatis* revealed by SPAAC or CuAAC.

DOI: https://doi.org/10.7554/eLife.37243.005

**Figure supplement 2.** Heterogeneous envelope probe labeling in *M. tuberculosis*.

*Figure 2 continued on next page*

*Figure 2 continued*

DOI: https://doi.org/10.7554/eLife.37243.006

**Figure supplement 2—source data 1.** Flow cytometry and conventional fluorescence microscopy images of *M. tuberculosis* labeled with different probes.

DOI: https://doi.org/10.7554/eLife.37243.007

is coincident. We then compared the fluorescence intensity profiles of cells that had been individually labeled with the probes, and found similar, average distributions of RADA, alkDADA and OalkTMM at the poles and peripheries of the cells (*Figure 2C*).

We next sought to address whether the cell envelope of the related *M. tuberculosis* is also labeled in polar gradients. We previously showed that alkDA incorporates into the cell surface of the organism (*Siegrist et al., 2013*) but were unable to stain the entire population of bacteria. To investigate the origin of labeling heterogeneity, we first tested whether the structure of fluorophore (*Figure 1B*) influenced probe incorporation by incubating *M. tuberculosis* in HADA, NADA or RADA and assessing population fluorescence by flow cytometry. HADA and NADA incubation yielded well-defined fluorescent populations (*Figure 2—figure supplement 2A*). RADA also labeled the entire *M. tuberculosis* population, albeit with greater cell-to-cell variability in fluorescence intensity. Given that *M. tuberculosis* incorporates fluorescent probes HADA and NADA relatively evenly across the population, we hypothesized that the apparent heterogeneity that we previously observed for alkDA labeling (*Siegrist et al., 2013*) was the result of an inefficient CuAAC ligation. We obtained very modest improvements by changing the reaction conditions, more specifically, by swapping the BTTP ligand (*Yang et al., 2014*; *Besanceney-Webler et al., 2011*) for the TBTA ligand, altering the ratio of ligand: Cu(I) and increasing the azide label concentration. We also switched our detection moiety to an azide appended to hydroxycoumarin, the same small, uncharged fluorophore as the one-step HADA probe (*Figure 1B*). Under our optimized conditions we detected azido-coumarin fluorescence from ~5 to 10% of cells that had been incubated in alkDA, alkDADA or OalkTMM (*Figure 2—figure supplement 2B and C*).

Although unable to achieve homogenous *M. tuberculosis* labeling with two-step envelope probes, we decided to test whether the sites of envelope labeling in the limited fluorescent subpopulation resemble those of *M. smegmatis*. HADA, alkDADA and OalkTMM tagging all produced cells that had a mixture of sidewall and polar fluorescence (*Figure 2—figure supplement 2C*) but exhibited a higher degree of cell-to-cell variability compared to *M. smegmatis*. Quantitation of HADA fluorescence showed that peptidoglycan metabolism comprised asymmetric polar gradients when averaged across the population (*Figure 2D*). We next asked whether there was a cell-wide difference in labeling distribution in *M. tuberculosis* compared to *M. smegmatis*. We arbitrarily defined the dimmer polar region as the first 15% of the cell length, the sidewall as the middle 70%, and the brighter polar region as the final 15%. As HADA labeled a large proportion of *M. tuberculosis* (*Figure 2—figure supplement 2A*) and was more resistant to photobleaching than NADA, we compared these ratios for HADA in septating and non-septating *M. smegmatis* and *M. tuberculosis* labeled for ~10% generation time (*Figure 2D*; under our growth conditions, *M. smegmatis* and *M. tuberculosis* generation times are 2.5–3 hr and 18–20 hr, respectively). Approximately 70–75% of HADA labeling in *M. tuberculosis* localized to the sidewall compared to 63–64% for *M. smegmatis* (*Figure 2D*). Thus, a greater proportion of peptidoglycan metabolism in *M. tuberculosis* likely occurs along the cell periphery than in *M. smegmatis*, although we cannot rule out a differential contribution of cyan autofluorescence in the two species (*Patiño et al., 2008*).

## Intracellular and extracellular pathways of D-amino acid probe incorporation in mycobacteria

As the cell periphery is not known to support surface expansion in mycobacteria (*Aldridge et al., 2012*; *Santi et al., 2013*; *Meniche et al., 2014*; *Thanky et al., 2007*; *Kieser and Rubin, 2014*; *Singh et al., 2013*; *Joyce et al., 2012*), we sought to characterize the molecular processes that underlie D-amino acid labeling patterns. OalkTMM and NalkTMM are inserted directly by the extracellular Antigen 85 complex into the mycomembrane (*Foley et al., 2016*). However, there are three potential pathways by which D-amino acid probes might incorporate into mycobacterial

peptidoglycan (*Siegrist et al., 2015*; *Cava et al., 2011*; *Ngo et al., 2016*): an intracellular, biosynthetic route and two extracellular routes, mediated by D,D-transpeptidase or L,D-transpeptidase remodeling enzymes (*Figure 3A*). Peptidoglycan remodeling, particularly by the L,D-transpeptidases abundantly encoded in the mycobacterial genome, may not strictly correlate with synthesis of the biopolymer (*Brown et al., 2012*; *Kuru et al., 2012*; *de Pedro and Cava, 2015*; *Glauner and Höltje, 1990*). Therefore, we sought to distinguish the different routes of incorporation for D-amino acid probes in mycobacteria.

It seemed possible that the chemical structure of the derivative (*Figure 1B*) and/or number of labeling steps (*Figure 1A*) might influence probe uptake, so we first tested the labeling sensitivity of a panel of D-amino acid derivatives to antibiotics that inhibit potential incorporation routes (*Figure 3A*). We also assessed OalkTMM and the D-alanine-D-alanine dipeptide probe, the latter of which has been proposed to tag the peptidoglycan of other species via the cytoplasmic MurF ligase [*Figure 3A*, *Figure 3—figure supplement 1*, (*Liechti et al., 2014*; *Sarkar et al., 2016*)]. D-cycloserine is a cyclic analog of D-alanine that inhibits the D-alanine racemase (Alr) and ligase (DdlA) in mycobacteria (*Feng and Barletta, 2003*). Together with the β-lactamase inhibitor clavulanate, β-lactams like ampicillin block D,D-transpeptidases and D,D-carboxypeptidases. Broader spectrum carbapenems such as imipenem additionally inhibit L,D-transpeptidases (*Kumar et al., 2012*). We also included vancomycin, an antibiotic that interferes with transpeptidation and transglycosylation by steric occlusion, to control for general defects in periplasmic peptidoglycan assembly. We empirically determined a time frame of drug treatment that did not compromise *M. smegmatis* viability (*Figure 3—figure supplement 2*). Within this time frame, labeling for all single residue D-amino acid probes decreased significantly in response to imipenem or D-cycloserine treatment (*Figure 3B*). By contrast, labeling by the dipeptide probe was relatively resistant to these antibiotics. Neither vancomycin nor ampicillin had a clear effect on labeling by any of the probes, indicating that cell death, disruption of peptidoglycan polymerization, and abrogration of D,D-transpeptidation do not explain the metabolic incorporation differences. OalkTMM labeling was sensitive to D-cycloserine, but not to imipenem nor ampicillin, suggesting that transfer of mycolates to arabinogalactan may require peptidoglycan precursor synthesis but not transpeptidation.

To test whether distinct mechanisms of probe incorporation were inhibited by D-cycloserine and imipenem, we performed a chemical epistasis experiment. We first examined the effect of different drug concentrations on alkDA incorporation (*Figure 3C*). Treatment with either D-cycloserine or imipenem, but not ampicillin, resulted in dose-dependent inhibition of the probe-derived fluorescence that plateaued at approximately half of untreated levels. We then assessed the combined effect of D-cycloserine and imipenem by Bliss independence (*Bliss, 1956*), a commonly used reference model for predicting drug-drug interactions based on the dose response for the individual drugs. This method is most appropriate when dual inhibition proceeds via distinct mechanisms for example activity against different molecules, enzymes or pathways (*Fitzgerald et al., 2006*). The effects of D-cycloserine and imipenem on alkDA incorporation were the same or slightly greater than the potencies predicted by the antibiotics individually (*Figure 3D*). The primarily additive nature of these antibiotics in the Bliss independence model is consistent with the idea that D-cycloserine and imipenem block distinct pathways of alkDA incorporation, and therefore that the probe incorporates into mycobacterial peptidoglycan via both cytoplasmic and L,D-transpeptidase routes.

RADA and NADA labeling were more sensitive to imipenem than HADA and alkDA (*Figure 3B*), at multiple concentrations of drug (*Figure 3E*). To test whether the probes were differentially incorporated by L,D-transpeptidases, we knocked out three of the six enzymes encoded in the *M. smegmatis* genome. We chose to focus on LdtA, LdtB and LdtE because the *M. tuberculosis* homologs (Ldt$_{MT1}$, Ldt$_{MT2}$, and Ldt$_{MT4}$, respectively) have been shown in vitro to have both cross-linking and D-amino acid exchange activity and to be inhibited by imipenem (*Cordillot et al., 2013*). RADA and NADA fluorescence decreased by ~80–85% the absence of *ldtABE*, and this effect was complemented by the expression of *ldtA* alone (*Figure 3F*). We observed a more moderate effect on HADA and alkDA labeling, which were decreased by ~40–60%. Collectively these data suggest that: 1. RADA and NADA probes primarily report L,D-transpeptidase activity, tetrapeptide substrate, or both, and 2. HADA and alkDA likely label via both cytoplasmic and L,D-transpeptidase routes.

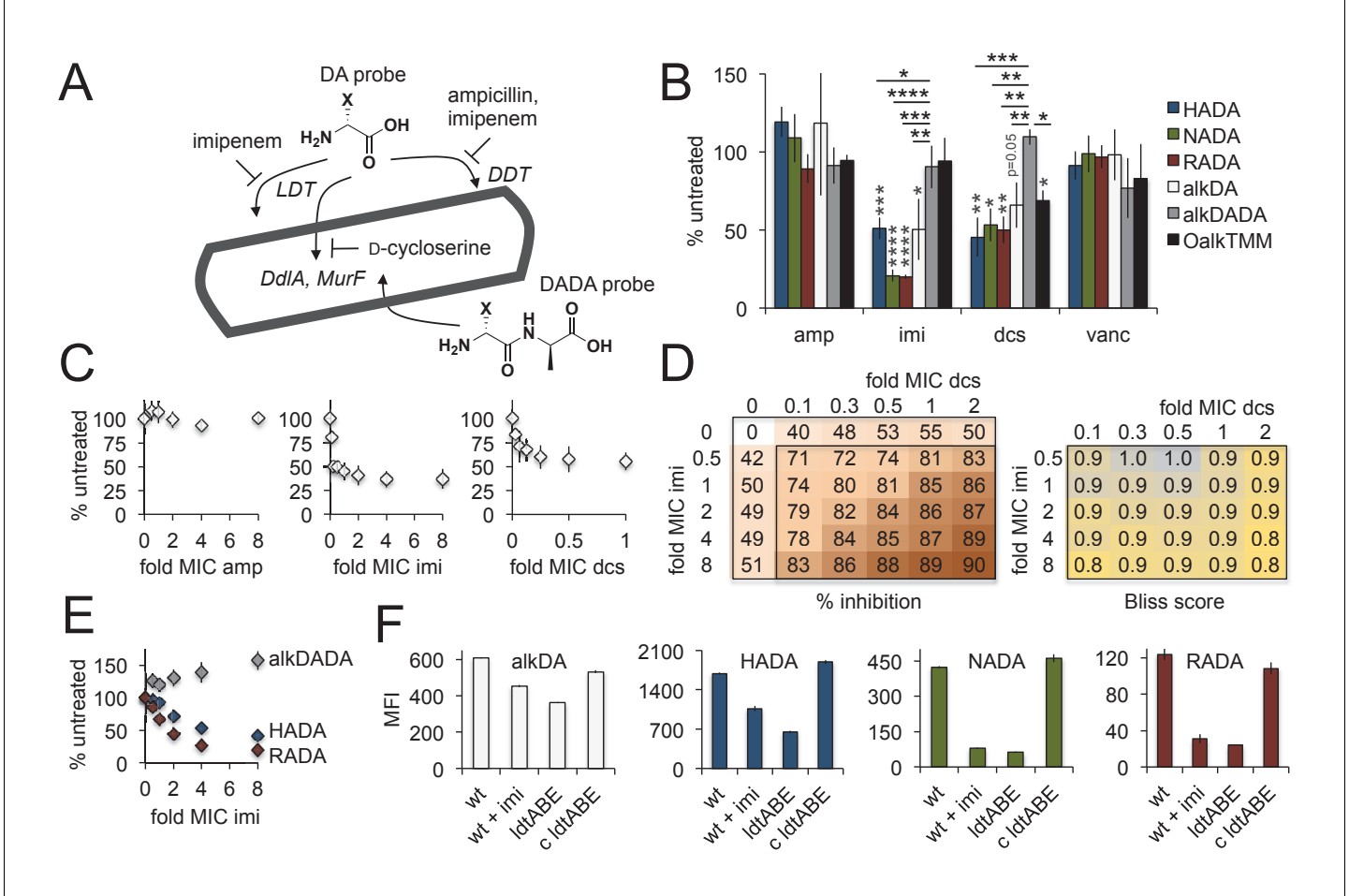

**Figure 3.** Multiple pathways of D-amino acid probe incorporation in *M. smegmatis*. (A) Schematic of the theoretical routes of D-amino acid (DA) and D-alanine D-alanine (DADA) probe incorporation. LDT, L,D-transeptidase, DDT, D,D-transeptidase (DDTs). For more details see *Figure 3—figure supplement 1*. (B) Sensitivity of HADA (blue), NADA (green), RADA (red), alkDA (light grey), alkDADA (dark grey) and OalkTMM (black) to antibiotics. Imi, imipenem + clavulanate; amp, ampicillin + clavulanate; dcs, D-cycloserine; vanc, vancomycin. *M. smegmatis* was pretreated or not with the indicated antibiotics at 2X MIC for 30 min then incubated an additional 15 min in the presence of probe. The bacteria were then washed and fixed. The alkyne-bearing probes were detected by CuAAC with azido-CR110 and quantitated by flow cytometry. Experiment was performed three to four times in triplicate. For each biological replicate, the averaged median fluorescence intensities (MFI) of the drug-treated samples were divided by the MFI of untreated bacteria. Data are expressed as the average percentage of untreated labeling across the biological replicates. Error bars, ±standard deviation. Statistical significance compared to alkDADA was assessed by two-tailed Student's t test of percentages for biological replicates. Horizontal black stars: *p<0.05; **p<0.005; ***p<0.0005; ****p<0.00005. Statistical significance compared to probe-matched controls that were not treated with antibiotic was assessed by two-tailed Student's t test of $\log_{10}$ MFI data for biological replicates. Vertical dark grey stars: *p<0.05; **p<0.005; ***p<0.0005; ****p<0.00005. (C) Effect of antibiotic dose on alkDA-derived fluorescence. *M. smegmatis* was pretreated or not with drugs at the fold-MIC indicated and labeled as in B. Experiment was performed three times in triplicate. For each biological replicate, the averaged MFI of the control (no drug, no alkDA but subjected to CuAAC) was subtracted from the averaged MFI of the drug-treated sample. This was then divided by the averaged MFI of untreated control (no drug but incubated in alkDA and subjected to CuAAC) from which the control MFI had also been subtracted. Data are expressed as the average percentage of untreated labeling across the biological replicates. Error bars, ±standard deviation. (D) Left, combined effects of imipenem and D-cycloserine on alkDA-derived fluorescence. *M. smegmatis* was pretreated or not with the drugs at the fold-MIC indicated and labeled as in B. Experiment was performed twice in triplicate with similar results. One data set is shown. The percent of untreated labeling was calculated as in C and subtracted from 100 to obtain the percent inhibition. Right, Bliss interaction scores for each pair of doses in left-hand graph were calculated as $(E_I + E_D E_I E_D)/E_{I,D}$ where $E_I$ is the effect of imipenem at dose *i*, $E_D$ is the effect of D-cycloserine at dose *d* and $E_{I,D}$ is the observed effect of the drugs at dose *i* and dose *d*. Combinations that produce Bliss scores greater than, equal to, or less than one are, respectively, interpreted as antagonistic, additive, or synergistic interactions. (E) Dose-dependent effect of imipenem on alkDADA (grey), HADA (blue) and RADA (red) labeling. *M. smegmatis* was pretreated or not with imipenem at the fold-MIC indicated and labeled as in B. Experiment was performed three times in triplicate and one representative data set is shown. For each technical replicate, the averaged median fluorescence intensities (MFI) of the drug-treated samples were divided by the averaged MFI of untreated bacteria. Data are expressed as the average percentage of untreated labeling across the technical replicates. Error bars, ±standard deviation. (F) Wildtype *M. smegmatis* pre-treated or not with 2X MIC imipenem and untreated Δ*ldtABE* and complement (c

*Figure 3 continued on next page*

*Figure 3 continued*

ldtABE) were labeled with alkDA (white), HADA (blue), NADA (green) or RADA (red) and processed as in B. Experiment was performed two to ten times in triplicate. Representative data from one of the biological replicates is shown here. Error bars, ±standard deviation.

DOI: https://doi.org/10.7554/eLife.37243.009

The following source data and figure supplements are available for figure 3:

**Source data 1.** Flow cytometry data of mycobacteria incorporating peptidoglycan probes in different conditions.
DOI: https://doi.org/10.7554/eLife.37243.013
**Figure supplement 1.** Schematic of peptidoglycan synthesis in mycobacteria.
DOI: https://doi.org/10.7554/eLife.37243.010
**Figure supplement 2.** Antibiotics do not cause obvious cell death in 45 min.
DOI: https://doi.org/10.7554/eLife.37243.012
**Figure supplement 2—source data 1.** Growth of *M. smegmatis* in presence of antibiotics.
DOI: https://doi.org/10.7554/eLife.37243.011

## Dipeptide D-amino acid probe incorporates into mycobacterial peptidoglycan precursors

As RADA labeling in mycobacteria is largely indicative of L,D-transpeptidase activity (*Figure 3*) and occurs at the poles and along the sidewall (*Figure 2*), we surmise that peptidoglycan is remodeled at both of these locations. We wished to determine whether remodeling was coincident with biopolymer synthesis but were limited by the multiple incorporation routes of alkDA and HADA (*Figure 3*). Dipeptide D-amino acid probes have been proposed to report peptidoglycan synthesis in other species via a cytoplasmic, MurF-dependent pathway [*Figure 3A*, *Figure 3—figure supplement 1*, (*Liechti et al., 2014*; *Sarkar et al., 2016*)]. Consistent with this notion, and in contrast to the monopeptide probes, alkDADA labeling was relatively stable to imipenem treatment (*Figure 3B and E*) and inefficiently incorporated (*Figure 4—figure supplement 1*). Thus we were surprised to observe that overall labeling by alkDADA decreased in the absence of LdtA, LdtB, LdtE or combinations thereof (*Figure 4—figure supplement 2A*). The reduction in signal occurred primarily at the poles (*Figure 4—figure supplement 2B*) yet loss of the enzymes did not impair bacterial growth (*Figure 4—figure supplement 3*). As we do not yet understand the mechanistic basis for this observation, we sought to directly test the hypothesis that dipeptide probes incorporate into peptidoglycan precursors.

D-alanine is produced in mycobacteria by Alr, the D-alanine racemase (*Figure 3—figure supplement 1*). The molecule is linked to a second D-alanine by DdlA, the D-alanine ligase, and the resulting dipeptide is added to the UDP-MurNAc-tripeptide by MurF. If the alkDADA probe is able to label mycobacterial peptidoglycan via MurF, addition of the molecule to the growth medium should rescue a mutant that is unable to make D-alanine-D-alanine. We first constructed an *alr* deletion mutant in *M. smegmatis* and verified that growth is rescued by exogenous D-alanine but not by alkDA (*Figure 4A*). Although our antibiotic data suggest that alkDA is incorporated into peptidoglycan in part via a cytoplasmic pathway (*Figure 3*), the inability of this probe to rescue growth was not surprising given the substrate specificities of Ddl and MurF (*Barreteau et al., 2008*) and inefficient synthesis of UDP-MurNAc-pentapeptide with D-amino acids other than alanine (*Cava et al., 2011*). In contrast to the alkDA results, we were able to rescue *alr* with either D-alanine-D-alanine or its alkynyl derivative (*Figure 4A*).

We considered the possibility that alkDADA may be digested by a D,D-carboxypeptidase prior to incorporation. This would result in the release of both unlabeled D-alanine and alkDA, the first of which could account for *alr* growth rescue (*Figure 4A*). We reasoned that a more precise gauge for MurF-dependent incorporation of the intact probe would be whether it could support the replication of a strain unable to ligate D-alanine to itself. Therefore, we confirmed that alkDADA rescues the growth of a temperature-sensitive *ddlA* mutant (*Belanger et al., 2000*) at the non-permissive temperature (*Figure 4B*).

Our genetic data supported a MurF-dependent pathway of alkDADA incorporation into peptidoglycan. If true, the probe should be present in precursors such as lipid I and lipid II (*Figure 3—figure supplement 1*). To test this hypothesis, we first optimized for mycobacteria a recently reported protocol for detecting lipid-linked precursors (*Qiao et al., 2014*; *Qiao et al., 2017*). We extracted

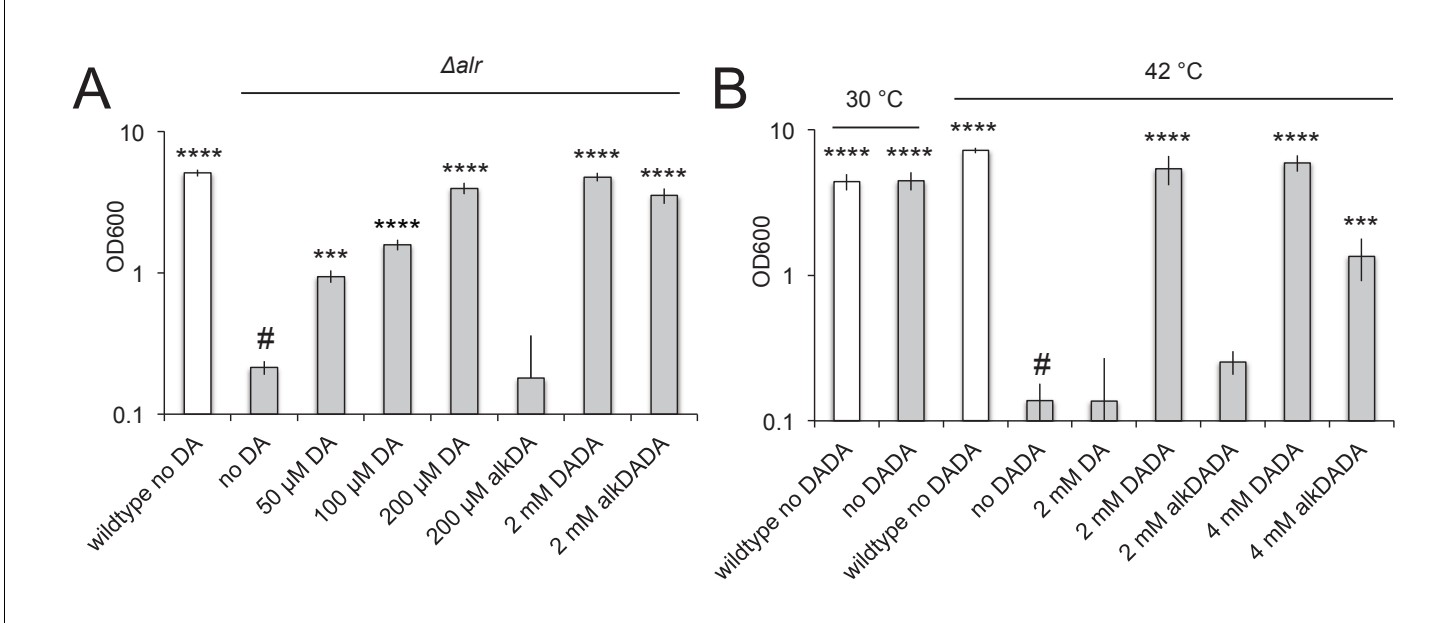

**Figure 4.** alkDADA rescues the growth of D-alanine racemase (Alr) and ligase (DdlA) mutants. alkDADA probe supports the growth of Δ*alr*, A, or temperature-sensitive *ddlA*ts, B. Wildtype (white bars) and Δ*alr* (grey bars), B, or *ddlA*ts (grey bars), were grown in the presence or absence of exogenous D-alanine, D-alanine- D-alanine or alkynyl derivatives thereof. Error bars, ±standard deviation. Significant differences compared to Δ*alr* no D-alanine (#, second bar from left), A, or *ddlA*ts at 42°C no D-alanine-D-alanine (#, fourth bar from left), B, One-way ANOVA with Dunnett's test, are shown for 3–4 biological replicates. ***p<0.005, ****p<0.0005.

DOI: https://doi.org/10.7554/eLife.37243.014

The following source data and figure supplements are available for figure 4:

**Source data 1.** OD$_{600}$ measurements of growth in different mycobacterial strains and conditions.
DOI: https://doi.org/10.7554/eLife.37243.020

**Figure supplement 1.** alkDA labels at much lower concentrations than alkDADA.
DOI: https://doi.org/10.7554/eLife.37243.015

**Figure supplement 1—source data 1.** Flow cytometry data of *M. smegmatis* incorporating AlkDa and AlkDADA.
DOI: https://doi.org/10.7554/eLife.37243.016

**Figure supplement 2.** Loss of LdtA, LdtB and/or LdtE decreases both RADA and alkDADA labeling.
DOI: https://doi.org/10.7554/eLife.37243.017

**Figure supplement 2—source data 1.** Flow cytometry and conventional fluorescence images of Ldt mutants labeled with peptidoglycan probes.
DOI: https://doi.org/10.7554/eLife.37243.018

**Figure supplement 3.** Loss of LdtA, LdtB and LdtE do not cause a growth defect in *M. smegmatis.*
DOI: https://doi.org/10.7554/eLife.37243.019

lipidic species from *M. smegmatis* and exchanged endogenous D-alanines for biotin-D-lysine (BDL) in vitro using purified *Staphylococcus aureus* enzyme PBP4, a promiscuous D,D-transpeptidase (*Qiao et al., 2014*). Biotinylated species were separated by SDS-PAGE and detected by horseradish peroxidase-conjugated streptavidin. We detected a biotin-linked, low-molecular-weight band that is reduced upon D-cycloserine treatment and accumulates when the lipid II flippase MurJ (MviN, (*Gee et al., 2012*)) is depleted or vancomycin is added (*Figure 5A*), conditions that have been shown to dramatically enhance precursor detection in other species (*Qiao et al., 2014*; *Qiao et al., 2017*). These data strongly suggest that the BDL-marked species are lipid-linked peptidoglycan precursors. We next turned our attention to detecting lipid I/II from *M. smegmatis* incubated with alkDADA. Initially we were unable to identify alkDADA-labeled species from organic extracts of wild-type *M. smegmatis* that had been subjected to CuAAC ligation with picolyl azide biotin (*Figure 5B*). We reasoned that the proportion of labeled precursors might be below our limit of detection. Accordingly, we repeated the experiment in the Δ*alr* background and found that we could clearly detect a low molecular-weight species band that accumulated with vancomycin treatment (*Figure 5B*, top) and that ran at the same size as BDL-labeled material (*Figure 5B*, bottom). We

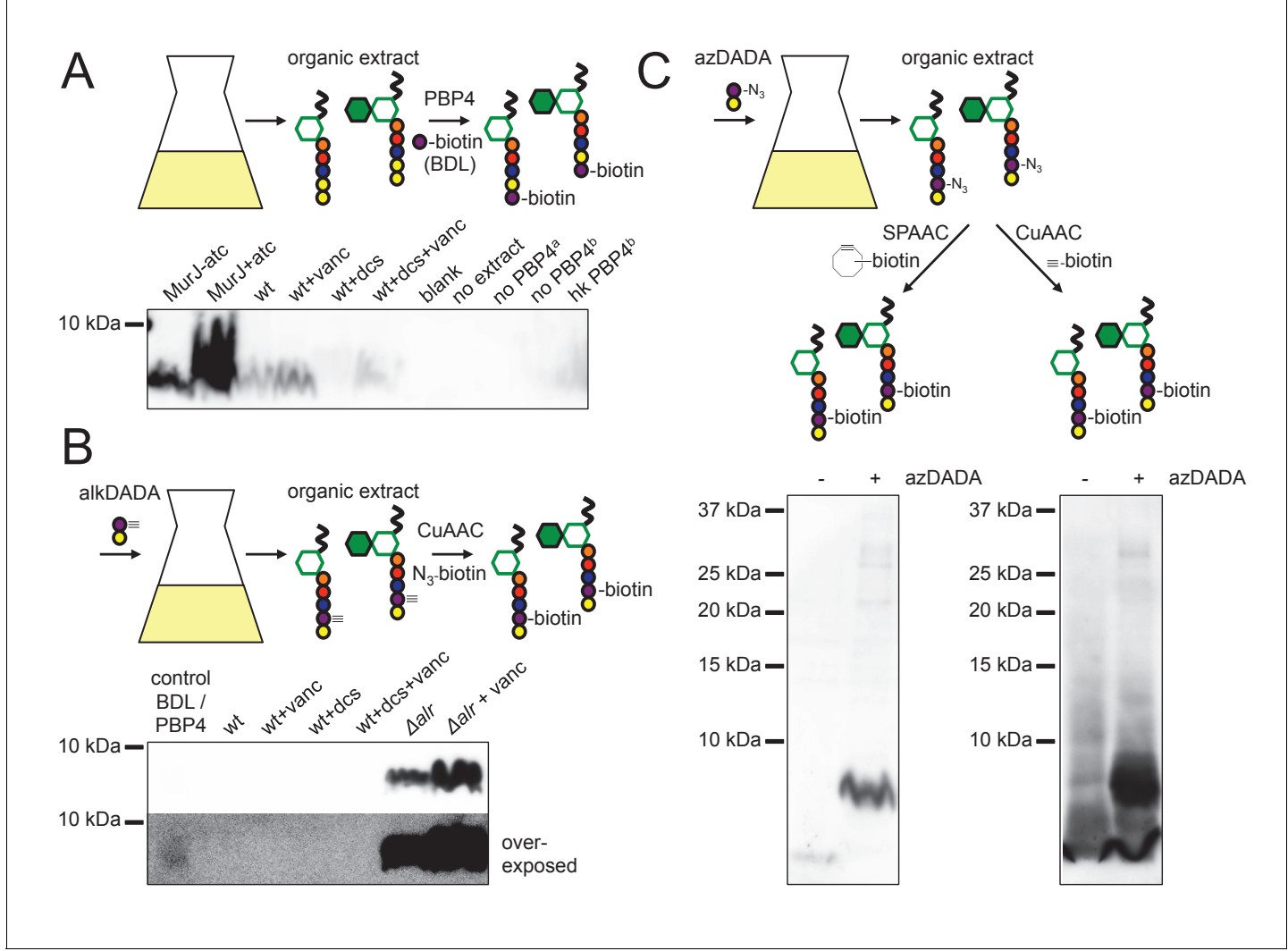

**Figure 5.** alkDADA and azDADA incorporate into lipid-linked peptidoglycan precursors. (**A**) Detection of lipid-linked peptidoglycan precursors from organic extracts of *M. smegmatis.* Endogenous D-alanines (yellow) were exchanged for biotin-D-lysine (BDL; purple) via purified *S. aureus* PBP4. Biotinylated species detected by blotting with streptavidin-HRP. MurJ (MviN) depletion strain was incubated in anhydrotetracycline (atc) to induce protein degradation. Other strains were treated with vancomycin (vanc), D-cycloserine (dcs) or a combination prior to harvesting. Wt, wildtype; blank, no sample run; no extract, BDL and PBP4 alone [a], organic extract from MurJ - atc; [b], organic extract from MurJ + atc; hk, heat-killed. (**B**) Detection of lipid-linked peptidoglycan precursors labeled by alkDADA in vivo. Wildtype and Δ*alr* strains were incubated in alkDADA (purple and yellow) and treated or not with the indicated antibiotics prior to harvest. Alkyne-tagged species from organic extracts were ligated to picolyl azide biotin via CuAAC then detected as in A. BDL/PBP4, endogenous precursors from MurJ + atc were subjected to in vitro exchange reaction as in A. (**C**) Detection of lipid-linked peptidoglycan precursors labeled by azDADA in vivo. Δ*alr* was incubated in azDADA (purple and yellow). Azide-tagged species from organic extracts were ligated to DIFO-biotin via SPAAC, left, or to alkyne biotin via CuAAC, right, then detected as in A.

DOI: https://doi.org/10.7554/eLife.37243.021

The following source data and figure supplements are available for figure 5:

**Source data 1.** Blots showing dipeptide probes in endogenous mycobacterial peptidoglycan precursors.
DOI: https://doi.org/10.7554/eLife.37243.030

**Figure supplement 1.** Short incubation in alkDADA results in polar and sidewall labeling.
DOI: https://doi.org/10.7554/eLife.37243.022

**Figure supplement 1—source data 1.** Structured illumination microscopy images of *M. smegmatis* labeled with AlkDADA.
DOI: https://doi.org/10.7554/eLife.37243.023

**Figure supplement 2.** Fluorescent vancomycin (vanc-fl) labeling at poles and sidewall.
DOI: https://doi.org/10.7554/eLife.37243.027

**Figure supplement 2—source data 1.** Conventional fluorescence microscopy images of *M. smegmatis* labeled with fluorescent vancomycin.

*Figure 5 continued on next page*

*Figure 5 continued*

DOI: https://doi.org/10.7554/eLife.37243.024

**Figure supplement 3.** Penicillin-binding proteins are present along mycobacterial cell periphery.

DOI: https://doi.org/10.7554/eLife.37243.028

**Figure supplement 3—source data 1.** Structured illumination microscopy images of PonA1-mRFP and conventional fluorescence microscopy images of *M. smegmatis* labeled with Bocillin-Fl.

DOI: https://doi.org/10.7554/eLife.37243.025

**Figure supplement 4.** Physical expansion of the mycobacterial cell is confined to the poles and occurs more rapidly at the RADA-bright tip.

DOI: https://doi.org/10.7554/eLife.37243.029

**Figure supplement 4—source data 1.** Pulse-chase of *M. smegmatis* labeled with RADA.

DOI: https://doi.org/10.7554/eLife.37243.026

were also able to identify a low-molecular-weight species from Δ*alr* incubated with azDADA that was revealed by either CuAAC or SPAAC ligation to alkyne- or cyclooctyne-biotin, respectively (*Figure 5C*). Taken together, our genetic and biochemical experiments show that the alkDADA and azDADA probes insert into mycobacterial peptidoglycan precursors by a MurF-dependent route.

## Fluorescent vancomycin and penicillin-binding proteins localize to the poles and sidewall in mycobacteria

The final, lipid-linked peptidoglycan precursor lipid II is synthesized by MurG on the cytoplasmic side of the plasma membrane then flipped to the periplasm and polymerized [*Figure 3—figure supplement 1*, (*Zhao et al., 2017*)]. We previously showed that MurG fused to two different fluorescent proteins and expressed under two different promoters is present at both the poles and periphery of *M. smegmatis* (*Meniche et al., 2014*). Labeling by alkDADA marks similar subcellular locations even with pulses as short as ~1% generation time (*Figure 5—figure supplement 1*). These data suggest that lipid-linked peptidoglycan precursors are synthesized at lateral sites in addition to their expected localization at the poles. However, our standard experimental protocol for detecting envelope labeling is to perform CuAAC on fixed cells. Because formaldehyde fixation can permeabilize the plasma membrane to small molecules, labeled material may be intracellular, extracellular or both. Dipeptide labeling could therefore read out lipid I/II on the cytoplasmic face of the plasma membrane, uncrosslinked lipid II on the periplasmic side, or polymerized peptidoglycan.

To shed light on the potential fate(s) of peptidoglycan precursors made at different subcellular sites, we first stained live mycobacterial cells with fluorescent vancomycin. This reagent binds uncrosslinked peptidoglycan pentapeptides and does not normally cross the plasma membrane. Pentapeptide monomers are a low abundant species in *M. tuberculosis*, *M. abscessus* and *M. leprae* peptidoglycan (*Kumar et al., 2012*; *Mahapatra et al., 2008*; *Lavollay et al., 2008*; *Lavollay et al., 2011*), suggesting that fluorescent vancomycin primarily reports extracellular, lipid-linked precursors in this genus. Labeling of *M. smegmatis* with this probe revealed both polar and lateral patches (*Figure 5—figure supplement 2*) as previously noted (*Singh et al., 2013*). This observation suggests that at least some of the peptidoglycan precursors present along the periphery of the mycobacterial cell are flipped to the periplasm.

We next sought to address whether these molecules could be used to build the peptidoglycan polymer. Transglycosylases from both the PBP (penicillin-binding proteins) and SEDS (shape, elongation, division, and sporulation) families stitch peptidoglycan precursors into the existing meshwork (*Figure 3—figure supplement 1*, [*Boutte et al., 2016*; *Hayashi et al., 2016*; *Zhao et al., 2017*; *Cho et al., 2016*; *Meeske et al., 2016*; *Leclercq et al., 2017*; *Arora et al., 2018*]). If peptidoglycan precursors are polymerized along the lateral surface of the mycobacterial cell, at least a subset of these periplasmic enzymes must be present at the sidewall to assemble the biopolymer. Two conserved PBPs in mycobacteria are likely responsible for most of the peptidoglycan polymerization required for cell viability, PonA1 and PonA2 (7, 19, 59). Published images of PonA1-mRFP and PonA1-mCherry localization suggested that the fusion proteins might decorate the mycobacterial sidewall in addition to the cell tips (*Joyce et al., 2012*; *Hett et al., 2010*; *Kieser et al., 2015a*), but the resolution of the micrographs did not allow for definitive assignment. Therefore, we first verified the localization of PonA1-mRFP. We found that a subset of this fusion protein indeed homes to the lateral cell surface (*Figure 5—figure supplement 3A*).

We were concerned that overexpression of PonA1-mRFP causes aberrant polar morphology and is toxic to *M. smegmatis* (*Hett et al., 2010*; *Kieser et al., 2015a*) and about the propensity of mCherry to cluster (*Landgraf et al., 2012*). Because our attempts to produce PonA1 fusions with different fluorescent proteins were unsuccessful, we opted to take a complementary, activity-based approach. Fluorescent derivatives of β-lactam antibiotics bind specifically and covalently to PBPs, and therefore have been used to image active enzyme in both protein gels and intact cells (*Kocaoglu and Carlson, 2013*). Our images of whole cells labeled with Bocillin, a BODIPY conjugate of penicillin, were in agreement with those from a previous publication (*Plocinski et al., 2011*), and seemed to indicate that Bocillin binds both the poles and sidewall of *M. smegmatis* (*Figure 5—figure supplement 3B and C*). However, given the hydrophobicity of the BODIPY dye, we considered the possibility that Bocillin might nonspecifically associate with the greasy mycomembrane. Fluorescence across the cell surface was diminished by pre-treating cells with the β-lactam ampicillin, which prevents peptidoglycan assembly by binding to PBPs, but not D-cycloserine, which inhibits peptidoglycan synthesis in a PBP-independent manner (*Figure 5—figure supplement 3B and C*). These experiments suggest that at least some of the sidewall labeling of Bocillin is specific, and therefore, that PBPs are present and active in these locations.

## Expansion of the mycobacterial envelope is concentrated at the poles

Our data indicate that peptidoglycan precursors are made and likely polymerized both at the poles and sidewall. Peptidoglycan synthesis is often presumed to mark sites of bacterial cell growth. However, dispersed elongation has not been reported in mycobacteria. Accordingly we performed a pulse chase experiment to test whether cell expansion correlates with sites of metabolic labeling. After marking peptidoglycan with RADA, we tracked labeled and unlabeled cell surface during 15 min (~10% generation time) of outgrowth. While we cannot rule out sidewall expansion below our limit of detection, the fluorescence dilution in this experiment was consistent with previous reports (*Aldridge et al., 2012*; *Santi et al., 2013*; *Meniche et al., 2014*; *Kieser and Rubin, 2014*; *Boutte et al., 2016*; *Rego et al., 2017*) and restricted to the mycobacterial poles (*Figure 5—figure supplement 4*).

## Muramidase treatment increases peptidoglycan synthesis along the sidewall

What is the function of peptidoglycan assembly that does not directly contribute to physical expansion of the cell? We hypothesized that one role of growth-independent cell wall synthesis might be repair. More specifically, we reasoned that insertion of peptidoglycan building blocks directly along the cell periphery would enable a real-time, comprehensive response to damage (*Figure 6A*). Cell wall repair that is restricted to sites of mycobacterial growth, by contrast, would be confined to the poles and renew the cell surface only after several generations. Extended incubation of *M. smegmatis* (~48 hr) with the peptidoglycan-degrading enzyme lysozyme substantially decreases colony-forming units (*Kanetsuna, 1980*). We have also shown that spheroplasts generated by combined glycine and lysozyme treatment lack peptidoglycan (*Melzer et al., 2018*). Together these data indicate that the enzyme is able to access and damage peptidoglycan in intact cells. We challenged *M. smegmatis* for 30 min in a mixture of lysozyme and mutanolysin, another enzyme that has been extensively used for in vitro digestion of peptidoglycan. After washing away the enzyme, we assessed the sites of peptidoglycan synthesis by alkDADA labeling. Pre-treatment by the muramidases clearly shifted the fluorescence from the brighter pole towards the sidewall (*Figure 6B*). These data indicate that mycobacteria reallocate peptidoglycan assembly away from the faster growing pole and toward the periphery upon damage to the cell wall (*Figure 6A*).

## Discussion

In this work, we aimed to address the seemingly discrepant observations that, on the one hand, mycobacteria expand from their tips (*Figure 5—figure supplement 4*, [*Aldridge et al., 2012*; *Santi et al., 2013*; *Meniche et al., 2014*; *Kieser and Rubin, 2014*; *Boutte et al., 2016*]), and on the other, metabolically labeled cell wall and synthetic enzymes are detectable at both the poles and along the sidewall (*Figure 5—figure supplement 3* [*Meniche et al., 2014*; *Joyce et al., 2012*; *Hett et al., 2010*; *Kieser et al., 2015a*; *Plocinski et al., 2011*]). The first step to resolving this

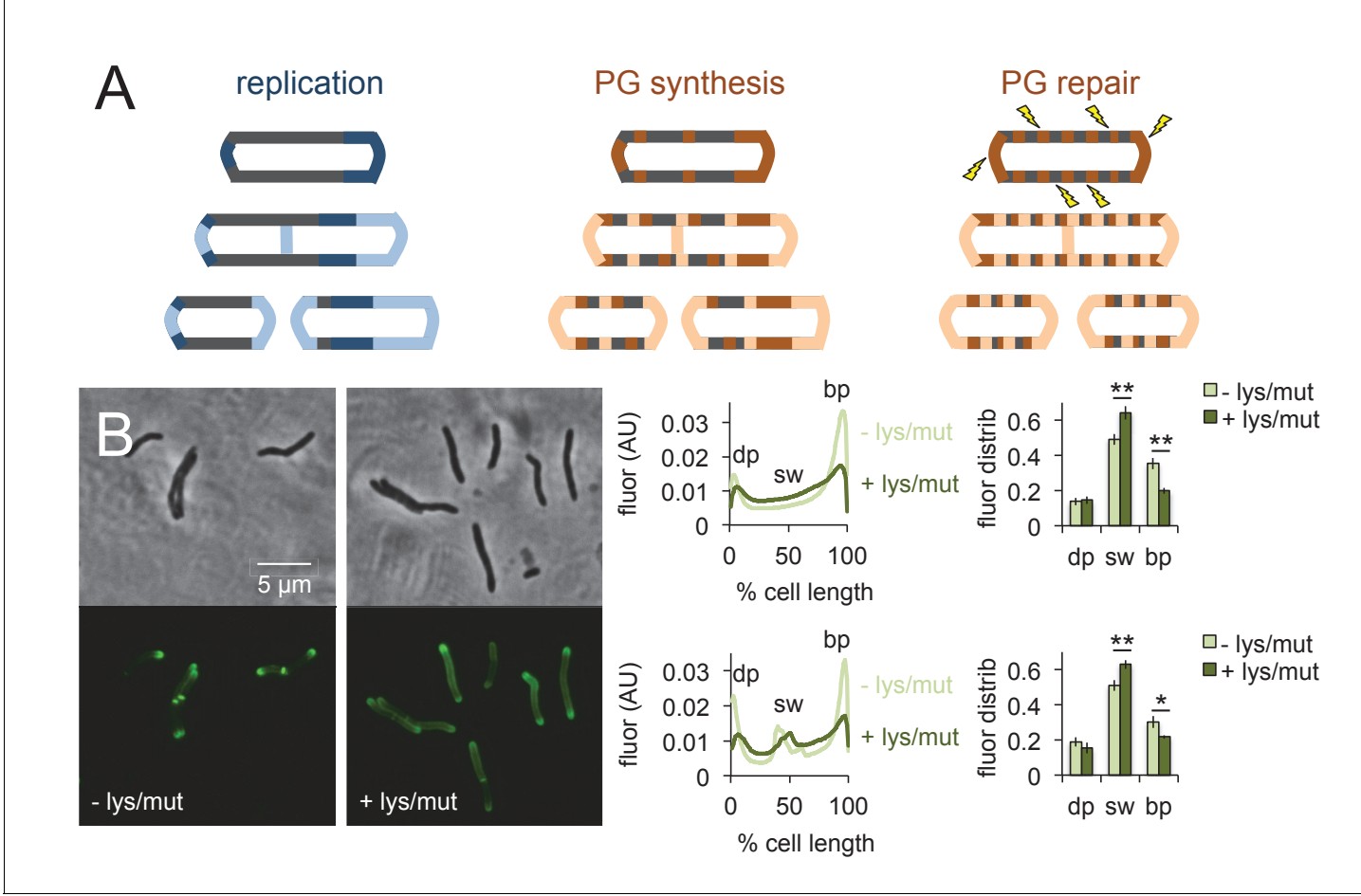

**Figure 6.** Peptidoglycan synthesis is redistributed to the sidewall upon cell wall damage. (**A**) Model for the spatial organization of peptidoglycan (PG) synthesis and repair with respect to mycobacterial growth and division. Left, regions of cell surface that expand are highlighted in blue. Middle and right, areas of peptidoglycan precursor synthesis are highlighted in orange. (**B**) *M. smegmatis* was pretreated (dark green) or not (light green) with lysozyme (lys) and mutanolysin (mut) for 30 min then incubated an additional 15 min in the presence of alkDADA. The bacteria were then washed and fixed and subjected to CuAAC with azido-CR110. Fluorescence was quantitated as in *Figure 2* for cells without (top; 232 < n < 236) and with (bottom; 29 < n < 55) visible septa. Dim pole (dp), sidewall (sw) and bright pole (bp) defined as in *Figure 2*. Fluor distrib, average fluorescence distribution. AU, arbitrary units. Error bars, ±standard deviation. Statistical significance between untreated (four biological replicates) and lysozyme/mutanolysin-treated (three biological replicates) was assessed for the dim pole, sidewall and bright pole by two-tailed Student's t test. *p<0.05; **p<0.005.
DOI: https://doi.org/10.7554/eLife.37243.031

The following source data is available for figure 6:

**Source data 1.** Conventional fluorescence microscopy images of *M. smegmatis* challenged with peptidoglycan-digesting enzymes.
DOI: https://doi.org/10.7554/eLife.37243.032

conundrum was to unambiguously identify sites of peptidoglycan synthesis. Although the D-amino acid probes that we and others have developed for peptidoglycan labeling have been extensively used for marking the cell wall (*Siegrist et al., 2015*), in most cases it has not been clear whether they report the location(s) of cytoplasmic synthesis, periplasmic exchange, or a combination of processes. Here, we show that in *M. smegmatis* the metabolic fate of monopeptide probes is partially dependent on the substituent; whereas NBD and TAMRA-conjugated D-amino acids are primarily exchanged into mycobacterial peptidoglycan by L,D-transpeptidases, their alkyne and coumarin-conjugated counterparts appear to incorporate by both extracellular and intracellular pathways (*Figure 3*). In the future, biochemical analysis of peptidoglycan composition will allow better quantitation of probe incorporation via different uptake pathways (*Cava et al., 2011*).

We show that dipeptide probes rescue the growth of a DdlA mutant (*Figure 4B*) and incorporate into lipid-linked peptidoglycan precursors (*Figure 5B and C*). To our knowledge, this is the first direct demonstration that peptidoglycan precursors can be metabolically labeled in vivo without radioactivity. Labeling by alkDADA unexpectedly decreased in the absence of L,D-transpeptidases (*Figure 4—figure supplement 2*). Unlike the monopeptide probes, however, alkDADA-derived fluorescence was stable to pre-treatment with imipenem, an antibiotic that targets this class of enzymes (*Figure 3*, [*Kumar et al., 2012*]). These data suggest that the dipeptide is unlikely to be a direct substrate for L,D-transpeptidases. It is possible that D,D-carboxypeptidases cleave a small proportion of alkDADA prior to incorporation and release D-alanine and alkDA. In this scenario, apparent alkDADA labeling in *M. smegmatis* may be a combination of intracellular alkDADA incorporation via MurF (*Figures 4* and *5*), extracellular alkDA incorporation by L,D-transpeptidases (*Figure 3*), and more limited intracellular alkDA incorporation by DdlA/MurF (*Figure 3*). This model is consistent with the promiscuous targeting of mycobacterial D,D-carboxypeptidases by carbapenems (*Kumar et al., 2012*) and the ~2000 fold more efficient incorporation of alkDA compared to alkDADA (*Figure 4—figure supplement 1*). Alternatively or additionally, alkDADA may be degraded abiotically or by other enzymes such as L,D-carboxypeptidases. Finally, loss of L,D-transpeptidases may introduce global alterations in peptidoglycan metabolism. Importantly, despite differences in overall fluorescence between dipeptide-labeled wild-type and Δ*ldtABE* (*Figure 4—figure supplement 1A*), sidewall fluorescence was preserved in the mutant (*Figure 4—figure supplement 1B*). In aggregate our data strongly suggest that the mycobacterial periphery is a site of active peptidoglycan synthesis.

The lateral surface of mycobacteria does not appear to contribute to cell elongation under normal growth conditions (*Figure 5—figure supplement 4*, [*Aldridge et al., 2012*; *Santi et al., 2013*; *Meniche et al., 2014*; *Kieser and Rubin, 2014*; *Boutte et al., 2016*]) but nevertheless hosts a substantial portion of envelope synthesis and remodeling. The intracellular difference in signal between the poles reflected relative elongation rates, as the RADA-bright cell tip, which coincides with the alkDADA- and OalkTMM-bright cell tip (*Figure 2B*) grows faster than the RADA-dim cell tip (*Figure 5—figure supplement 4*). The twofold ratio of fast/bright:slow/dim pole fluorescence (*Figures 2C, D* and *6B*) roughly corresponds to previous estimates of intracellular differences in polar elongation (*Aldridge et al., 2012*; *Joyce et al., 2012*). Compared to *M. smegmatis*, the distribution of HADA labeling in *M. tuberculosis* is shifted away from the fast pole toward the periphery (*Figure 2D*). Diminished polarity and asymmetry is also apparent in the sub-population of *M. tuberculosis* that is labeled by alkDADA and OalkTMM (*Figure 2—figure supplement 2C*). Our data are in agreement with the heterogeneity in polar dominance observed previously for *M. tuberculosis* (*Botella et al., 2017*). Although we cannot rule out a contribution from cyan autofluorescence, these experiments also suggest that sidewall envelope metabolism may be even more prominent in *M. tuberculosis* than in *M. smegmatis*, comprising 70–75% of the total cell output.

It is possible that peptidoglycan assembly along the lateral surface of the mycobacterial cell is simply a byproduct of synthetic enzymes that are en route to the polar elongasome or the divisome. Having active enzymes at the ready could enable efficient coordination between cell growth and septation. We think that this model is less likely, however, given the energetic cost of producing complex macromolecules and the known limits on the steady-state pools of lipid-linked peptidoglycan precursors (*van Heijenoort et al., 1992*). Instead we propose that cell wall synthesis along the periphery could allow mycobacteria to edit what would otherwise be an inert surface (*Figure 6A*). Peptidoglycan and mycomembrane metabolism in this region may thicken or fill in the gaps of envelope that was initially deposited at the poles or the septum and enable the bacterium to correct stochastic defects and repair damage. In support of this model, we find that cell wall synthesis along the sidewall is enhanced upon exposure to peptidoglycan-degrading enzymes (*Figure 6B*). More broadly, the ability to tailor the entire cell surface, not just the ends, should enable rapid adaptation to external stimuli. Such activity may be particularly important for *M. tuberculosis*, a slow-growing organism that must survive a hostile, nutrient-poor environment.

## Materials and methods

**Key resources table**

*Continued on next page*

*Continued*

| Reagent type (species) or resource | Desig-nation | Source or reference | Identifiers | Additional information |
|---|---|---|---|---|
| Reagent type (species) or resource | Desig-nation | Source or reference | Identifiers | Additional information |
| Strain (*M. smegmatis* mc2155) | *M. smegmatis* | NC_008596 in GenBank | | Wildtype *M. smegmatis* |
| Genetic reagent (*M. smegmatis*) | Δ*alr* | This paper | | The mutant was generated by recombineering protocols described in DOI: 10.1038/nmeth996 and DOI: 10.1007/978-1-4939-2450-9_10; see the methods section for further detail. |
| Genetic reagent (*M. smegmatis*) | Δ*ldtA* | doi:10.1101/291823 | | Obtained from Dr. Eric Rubin (Harvard SPH) and Dr. Hesper Rego (Yale Med) |
| Genetic reagent (*M. smegmatis*) | Δ*ldtB* | doi:10.1101/291823 | | Obtained from Dr. Eric Rubin (Harvard SPH) and Dr. Hesper Rego (Yale Med) |
| Genetic reagent (*M. smegmatis*) | Δ*ldtE* | doi:10.1101/291823 | | Obtained from Dr. Eric Rubin (Harvard SPH) and Dr. Hesper Rego (Yale Med) |
| Genetic reagent (*M. smegmatis*) | Δ*ldtAE* | doi:10.1101/291823 | | Obtained from Dr. Eric Rubin (Harvard SPH) and Dr. Hesper Rego (Yale Med) |
| Genetic reagent (*M. smegmatis*) | Δ*ldtBE* | doi:10.1101/291823 | | Obtained from Dr. Eric Rubin (Harvard SPH) and Dr. Hesper Rego (Yale Med) |
| Genetic reagent (*M. smegmatis*) | Δ*ldtBA* | doi:10.1101/291823 | | Obtained from Dr. Eric Rubin (Harvard SPH) and Dr. Hesper Rego (Yale Med) |
| Genetic reagent (*M. smegmatis*) | Δ*ldtABE* | doi:10.1101/291823 | | Obtained from Dr. Eric Rubin (Harvard SPH) and Dr. Hesper Rego (Yale Med) |
| Genetic reagent (*M. smegmatis*) | MurJ (MviN) | doi: 10.1126/scisignal. 2002525. | | Obtained from Dr. Chris Sassetti (U Mass Med) |
| Genetic reagent (*M. smegmatis*) | *ddlAts* | doi: 10.1128/JB.182.23. 6854–6856.2000 | | Obtained from Dr. Graham Hatfull (U Pitt) |
| Genetic reagent (*M. smegmatis*) | pTetO*ldtA* | doi: 10.1101/291823 | | Obtained from Dr. Eric Rubin (Harvard SPH) and Dr. Hesper Rego (Yale Med) |
| Genetic reagent (*M. tuberculosis*) | Δ*RD1* Δ*panCD* | doi: 10.1016/j.vaccine. 2006.05.097 | | Obtained from Dr. Bill Jacobs (Einstein Med) |
| Other | RADA | doi: 10.1002/anie.201206749; doi: 10.1038/nprot.2014.197 | | Synthesized by Tocris Bioscience (Bristol, United Kingdom) following referenced protocols |
| Other | NADA | doi: 10.1002/anie.201206749; doi: 10.1038/nprot.2014.197 | | Synthesized by Tocris Bioscience (Bristol, United Kingdom) following referenced protocols |
| Other | HADA | doi: 10.1002/anie.201206749; doi: 10.1038/nprot.2014.197 | | Synthesized by Tocris Bioscience (Bristol, United Kingdom) following referenced protocols |
| Other | alkyne-D-alanine (alkDA;EDA) | Thermo Fisher, Waltham, MA | Cat # AC441225000 | |
| Other | alkyne-D-alanine-D-alanine (alkDADA; EDADA) | doi: 10.1038/nature12892 | | Synthesized by the Chemical Synthesis Core Facility at Albert Einstein College of Medicine (NY, USA) following the referenced protocols |
| Other | azido-D-alanine-D-alanine (azDADA; ADADA) | doi: 10.1038/nature12892 | | Synthesized by the Chemical Synthesis Core Facility at Albert Einstein College of Medicine (NY, USA) following the referenced protocols |
| Other | O-alkyne-trehalose monomycolate (OalkTMM) | doi: 10.1002/anie.201509216 | | |
| Other | N-alkyne-trehalose monomycolate (NalkTMM) | doi: 10.1002/anie.201509216 | | |
| Software, algorithm | MATLAB codes | This paper | | Scripts designed for MATLAB to analyze the fluorescence profiles along a cell body from data collected in Oufti (doi: 10.1111/mmi.13264). |

*Continued on next page*

*Continued*

| Reagent type (species) or resource | Desig-nation | Source or reference | Identifiers | Additional information |
|---|---|---|---|---|
| Chemical compound, drug | Fmoc-D-Lys(biotinyl)-OH (BDL precursor) | Chem-Impex International | Cat # 16192 | Deprotected as described in doi: 10.1021/ja508147s to yield BDL |
| DNA reagent | PBP4 plasmid | doi: 10.1021/ja508147s | | Obtained from Dr. Suzanne Walker (Harvard Med) |

## Bacterial strains and culture conditions

mc$^2$155 *M. smegmatis* and Δ*RD1* Δ*panCD M. tuberculosis* (*Sambandamurthy et al., 2006*) were grown at 37°C in Middlebrook 7H9 growth medium (BD Difco, Franklin Lakes, NJ) supplemented with glycerol, Tween 80 and ADC (*M. smegmatis*) or OADC and 50 µg/ml pantothenic acid (*M. tuberculosis*). The Δ*alr* strain was further supplemented with 1 mM D-alanine. The *ddlA*$^{ts}$ strain was grown in 7H9-ADC at 30°C or 37°C as specified.

## Cell envelope labeling

Probes used in this study include (i) fluorescent D-amino acids HADA, NADA and RADA (Tocris Bioscience, Bristol, UK) (ii) alkDA (R-propargylglycine, Thermo Fisher, Waltham, MA) (iii) alkDADA and azDADA (Chemical Synthesis Core Facility, Albert Einstein College of Medicine, Bronx, NY) and (iv) OalkTMM and NalkTMM, synthesized as described previously (*Foley et al., 2016*). The labeling procedures were performed with modifications from (*Kuru et al., 2012*; *Siegrist et al., 2013*; *Foley et al., 2016*). Unless otherwise indicated, mid-log *M. smegmatis* or *M. tuberculosis* were labeled with 500 µM HADA, 25 µM NADA or RADA, 50 µM alkDA, 1 or 2 mM alkDADA, 50 µM OalkTMM or 250 µM NalkTMM for 15 min or 2 hr, respectively. In some cases *M. smegmatis* was preincubated ± antibiotics at different fold-MIC (MICs = 80 µg/mL D-cycloserine, 8 µg/mL ampicillin (with 5 µg/mL clavulanate), 0.5 µg/mL imipenem (with 5 µg/mL clavulanate), 6 µg/mL vancomycin) or ± 500 µg/mL lysozyme and 500 U/mL mutanolysin for 30 min then grown for an additional 15 min in the presence of the probes. Cultures were then centrifuged at 4°C, washed in pre-chilled PBS containing 0.05% Tween 80% and 0.01% BSA (PBSTB) and fixed for 10 min in 2% formaldehyde at room temperature (RT). After two washes in PBSTB, the bacterial pellets were resuspended in half of the original volume of freshly prepared CuAAC solution in PBSTB (*Siegrist et al., 2013*) containing either AF488 picolyl azide or carboxyrhodamine 110 (CR110) azide (Click Chemistry Tools, Scottsdale, AZ) or 3-azido-7-c (Jena Biosciences, Jena, Germany). For *M. tuberculosis* experiments and in *Figure 2—figure supplement 1A*, we used a modified, low-copper CuAAC reaction protocol: 200 µM CuSO$_4$ and 800 µM BTTP (Chemical Synthesis Core Facility, Albert Einstein College of Medicine, Bronx, NY) were pre-mixed then added to PBSTB. Immediately before resuspending the pellets, 2.5 mM freshly prepared sodium ascorbate and 300 µM 3-azido-7-hydroxycoumarin were added to the mixture. After 30–60 min gentle agitation at RT, cultures were washed once with PBSTB, once with PBS and resuspended in PBS for imaging or flow cytometry analysis (FITC, BV510, and Texas Red channels on a BD DUAL LSRFortessa, UMass Amherst Flow Cytometry Core Facility). NHS ester dye labeling and fluorescent vancomycin labeling were performed as described (*Aldridge et al., 2012*; *Thanky et al., 2007*). *M. tuberculosis* was post-fixed with 4% formaldehyde overnight at RT prior to removing from the biosafety cabinet. For Bocillin labeling experiments, 500 µL of mid-log *M. smegmatis* was washed once in PBST and resuspended in PBST containing 5 µg/mL clavulanate. Bacteria were then pre-incubated or not with 50 µg/mL ampicillin or D-cycloserine at RT with gentle agitation. After 15 min, 50 µg/mL Bocillin FL (Thermo Fisher) was added and cultures were incubated for an additional 30 min. They were then washed three times in PBST and imaged live on agar pads.

## Genetic manipulation

The Δ*alr M. smegmatis* strain was generated using standard recombineering methods (*van Kessel and Hatfull, 2007*; *Murphy et al., 2015*). 500 bp up- and downstream of *alr* were cloned on either side of the *hyg*$^R$ cassette flanked with *loxP* sites. After induction of *recET*, transformation and subsequent selection on hygromycin and 1 mM D-alanine, PCR was used to confirm the presence of the correct insert. Strains were then cured of the *hyg*$^R$ cassette by transformation with an episomal

plasmid carrying the Cre recombinase and a sucrose negative selection marker. Strains were cured of this plasmid by repeated passaging in the presence of sucrose.

*M. smegmatis* lacking *ldtA, ldtB* and/or *ldtE* were generously provided by Dr. Kasia Baranowski, Dr. Eric Rubin and Dr. Hesper Rego and are described in bioRxiv https://doi.org/10.1101/291823. Briefly, strains were constructed by recombineering to replace the endogenous copies with zeocin or hygromycin resistance cassettes flanked by *loxP* sites as previously described above (*Boutte et al., 2016*). Once the knock-outs were verified by PCR, the antibiotic resistance cassettes were removed by the expression of Cre recombinase. To complement Δ*ldtABE,* a copy of *ldtA* was under the constitutive TetO promoter on a kanamycin marked vector (CT94) that integrates at the L5 phage integration site of the chromosome.

## Microscopy

Fixed bacteria were imaged either by conventional fluorescence microscopy (Nikon Eclipse E600, Nikon Eclipse Ti or Zeiss Axioscope A1 with 100x objectives) or by structured illumination microscopy (Nikon SIM-E/A1R with SR Apo TIRF 100x objective).

## Microscopy analysis

Images were processed using FIJI (*Schindelin et al., 2012*) and cells were outlined and segmented using Oufti (*Paintdakhi et al., 2016*). Fluorescence signals of each cell were detected using Oufti and analyzed using custom-written MATLAB codes. The fluorescence intensities that we report here have been normalized by cell area. We distinguished septating from non-septating cells using the probe fluorescence profile along the long cell axis. We used the peakfinderprogram (*Sliusarenko et al., 2011*) to identify peaks in the labeling profile. Because our probes label both the cell poles as well as the septum, septating cells were those that had three peaks in their labeling profile, with the middle peak positioned between 30% and 70% along the normalized long cell axis. Non-septating cells were identified as having only two peaks.

## Detection of lipid-linked peptidoglycan precursors

To detect endogenous lipid-linked peptidoglycan precursors, we adopted the assay developed in (*Qiao et al., 2014*) with some modifications. *M. smegmatis* was inoculated in 100 mL of 7H9 medium and grown to mid-log phase at 37°C. Where applicable, MurJ was depleted by 8 hr of anhydrotetracycline-induced protein degradation as described (*Gee et al., 2012*). The bacteria were then divided into 25 mL cultures that were subjected or not to freshly prepared 80 µg/mL vancomycin and/or 10 µg/mL D-cycloserine. After 1 hr of incubation at 37°C, bacteria were collected by centrifugation and cell pellets were normalized by wet weight. 200–300 mg wet pellet was resuspended in 500 µL 1% glacial acetic acid in water. 500 µL of the resuspended pellet mixture was transferred into a vial containing 500 µL of chloroform and 1 mL of methanol and kept at room temperature (RT) for 1–2 hr with occasional vortexing. The mixture was then centrifuged at 21,000x g for 10 min at RT and the supernatant was transferred into a vial containing 500 µL of 1% glacial acetic acid in water and 500 µL chloroform, and vortexed for 1 min. After centrifugation at 900x g for 2 min at RT, three phases were distinguishable: aqueous, an interface, and organic. We collected the lipids from the organic phase and from the interface and concentrated it under nitrogen. Organic extracts were resuspended in 12 µL of DMSO and then incubated with purified *S. aureus* PBP4 and biotin-D-lysine (BDL; deprotected from Fmoc-D-Lys(biotinyl)-OH; Chem-Impex International) as described (*Qiao et al., 2014*). Upon completion of the BDL exchange reaction, 10 µL of 2X loading buffer was added to the vials. Contents were boiled at 95°C for 5 min then run on an 18% SDS polyacrylamide gel. Biotinylated species were transferred to a PVDF membrane, blotted with streptavidin-HRP (diluted 1:10,000, Thermo-Fisher) and visualized in an ImageQuant system (GE Healthcare).

To detect lipid-linked precursors that had been metabolically labeled with alkDADA or azDADA, growth of the Δ*alr* strain was initially supported by the inclusion of 2 mM D-alanine-D-alanine (Sigma-Aldrich) in the 7H9 medium. Δ*alr* bacteria were harvested and washed twice in sterile PBST prior to resuspension in 100 mL of pre-warmed medium. Both wildtype and Δ*alr M. smegmatis* were incubated in 0.5–1 mM of alkDADA or azDADA for 1 hr then harvested as described above. Organic extracts from metabolically labeled cultures were subjected to CuAAC reaction by adding, in order and in a non-stick vial: 2 µL of PBST, 1 µL of 5 mM CuSO$_4$, 1 µL of 20 mM BTTP, 1 µL of freshly-

prepared 50 mM sodium ascorbate, 3 µL of 10 mM picolyl azide biotin or alkyne biotin (Click Chemistry Tools), and 2 µL of the organic extract. The reaction was incubated for 1 hr at RT with gentle shaking prior to detecting as above.

## Acknowledgements

We are grateful to Dr. Kasia Baranowski, Dr. Eric Rubin and Dr. Hesper Rego for sharing L,D-transpeptidase mutants and for providing critical feedback. We thank Dr. Suzanne Walker and Dr. Kaitlin Schaefer for the *S. aureus pbp4* expression construct and for helpful technical advice; Dr. Steven Sandler, Dr. Peter Chien, Dr. James Chambers and Dr. Amy Burnside for microscopy and flow cytometry guidance; Ms. Sylvia Rivera for technical assistance; Dr. Krista Gile for guidance on statistical analysis. We also acknowledge Dr. Chris Sassetti for the MurJ depletion strain, Dr. Graham Hatfull for the *ddlA*$^{ts}$ mutant, Dr. William Jacobs for $\Delta RD1 \Delta panCD$ *M. tuberculosis* and Dr. Yves Brun, Dr. Erkin Kuru and Dr. Michael VanNieuwenhze for the initial supply of the alkDADA (EDA-DA) probe.

## Additional information

### Funding

| Funder | Grant reference number | Author |
|---|---|---|
| National Institutes of Health | New Innovator Award DP2 AI138238 | M Sloan Siegrist |
| National Science Foundation | CAREER 1654408 | Benjamin M Swarts |
| Simons Foundation | Life Sciences Research Foundation Fellowship | Hoong Chuin Lim |
| Research Corporation for Science Advancement | Cottrell College Science Award 22525 | Benjamin M Swarts |
| National Institutes of Health | U01CA221230 | M Sloan Siegrist |
| National Institutes of Health | Training Grant T32 GM008515 | Emily S Melzer |

The funders had no role in study design, data collection and interpretation, or the decision to submit the work for publication.

### Author contributions

Alam García-Heredia, Conceptualization, Data curation, Formal analysis, Validation, Investigation, Methodology, Writing—review and editing; Amol Arunrao Pohane, Conceptualization, Formal analysis, Supervision, Investigation, Writing—review and editing; Emily S Melzer, Formal analysis, Investigation, Methodology, Writing—review and editing; Caleb R Carr, Conceptualization, Formal analysis, Investigation, Writing—review and editing; Taylor J Fiolek, Sarah R Rundell, Jeffrey C Wagner, Resources, Methodology, Writing—review and editing; Hoong Chuin Lim, Software, Methodology, Writing—review and editing; Yasu S Morita, Conceptualization, Methodology, Writing—review and editing; Benjamin M Swarts, Conceptualization, Resources, Funding acquisition, Methodology; M Sloan Siegrist, Conceptualization, Resources, Data curation, Formal analysis, Supervision, Funding acquisition, Investigation, Methodology, Writing—original draft, Project administration, Writing—review and editing

### Author ORCIDs

Alam García-Heredia (iD) http://orcid.org/0000-0002-9573-4087
Amol Arunrao Pohane (iD) http://orcid.org/0000-0003-2574-0366
Yasu S Morita (iD) http://orcid.org/0000-0002-4514-9242
M Sloan Siegrist (iD) https://orcid.org/0000-0002-8232-3246

Decision letter and Author response
Decision letter https://doi.org/10.7554/eLife.37243.038
Author response https://doi.org/10.7554/eLife.37243.039

## Additional files

### Supplementary files
• Source code 1. MatLab code to plot the fluorescence distribution of mycobacterial cells.
DOI: https://doi.org/10.7554/eLife.37243.033

• Transparent reporting form
DOI: https://doi.org/10.7554/eLife.37243.034

### Data availability

The source data generated during this study have been included in the supporting files. These data have also been deposited to the Open Science Framework (osf.io/8ynhx)

The following dataset was generated:

| Author(s) | Year | Dataset title | Dataset URL | Database and Identifier |
|---|---|---|---|---|
| M Sloan Siegrist | 2018 | Peptidoglycan precursor synthesis along the sidewall of pole-growing mycobacteria | https://dx.doi.org/10.17605/OSF.IO/8YNHX | Open Science Framework, 10.17605/OSF.IO/8YNHX |

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
