## [Decision Letter]

Thank you for submitting your article "Sidewall cell envelope synthesis and remodeling in pole-growing mycobacteria" for consideration by *eLife*. Your article has been reviewed Gisela Storz as the Senior Editor, a Reviewing Editor, and three reviewers. The following individual involved in review of your submission has agreed to reveal her identity: Bree Aldridge (Reviewer #2).

The reviewers have discussed the reviews with one another and the Reviewing Editor has drafted this decision to help you prepare a revised submission.

Summary:

Current views of mycobacterial cell growth dictate that these organisms primarily expand and remodel their cell surfaces at the polar regions. This is facilitated by an elongasome complex, comprising a tropomyosin-like protein (DivIVA), which forms a multiprotein complex with cell wall biosynthetic enzymes to allow for insertion of new cell wall material in the sub-polar region of the cell tip. The assumption in this case has been that side wall remodeling does not occur and that this region of the cell wall is inert. The submission by Siegrist and colleagues undertakes a detailed investigation of whether remodeling of the peptidoglycan (PG) layer occurs along the lateral axis of the mycobacterial cell. They hypothesize that if side-wall PG is not inert, enzymes should be present to allow for remodeling in this region. To test this, they first localize PonA1 (a HMW PBP) in Mycobacterium smegmatis and show that notable side wall localization occurs, further confirmed by using a BODIPY conjugate of penicillin (bocillin), that should bind HMW PBPs. Side wall binding by bocillin was abrogated with ampicillin treatment, confirming the association of the conjugate with PBPs. The authors confirm these observations by using metabolic probes, which included derivatives of D-amino acids and report bidirectional gradients of fluorescence, emanating from the cell pole and weakened towards mid-cell. Similar patterns were observed with trehalose probes that label mycolic acids, confirming that biosynthesis of the various layers in the mycobacterial envelope are spatially coordinated. Next, the authors sought to test whether similar staining patterns prevail in *M. tuberculosis* however, uptake of the probes was poor, with a high degree of heterogeneity. After some optimization, the population of cells that stained had a mixture of polar and side wall labeling. By comparing the ratios of staining intensities at the pole versus the side-wall the authors suggest that in mycobacteria, a notable proportion of PG labeling by probes occurs on the sidewall. Using a pulse chase experiment with RADA, the authors demonstrate that fluorescence dilution is asymmetric. After this, the authors sought to assess how D-amino acid probes are incorporated into mycobacterial PG by selectively inhibiting cytoplasmic and periplasmic steps pf PG synthesis using antibiotics. La belling with single D-amino acids was reduced with antibiotics that target either the cytoplasmic or periplasmic incorporation of these probes. In contrast, labeling with the dipeptide (alkDADA) was unaffected. To test the role of L,D-transpeptidase (Ldts) in probe incorporation, the authors deleted three of the six putative Ldts in M smegmatis (ΔldtABE) and demonstrated a reduction in probe uptake. Next, the authors report that the dipeptide probe labelled cells with reduced efficiency when compared to the single amino acid forms. To further study this, the authors abrogate alanine racemase activity, which should eliminate de novo synthesis of the dipeptide. The resulting mutant could be rescued with exogenous D-ala-D-ala supplementation, confirming the role of MurF in incorporating this alkDADA probe, further evidenced by isolation of lipid-linked, labelled derivatives. Finally, the authors demonstrate that alkDADA labeling can be modulated by treatment with imipenem.

Central conclusions:

1) Notable sidewall remodeling of PG occurs in mycobacteria, linked to the activity of PBPs

2) Incorporation of the alanine derivatives, but not the dipeptide probe (alk-DADA) is inhibited by Imipenem

3) Alk-DADA can rescue the growth of a D-alanine racemase and ligase mutants

4) Alk-DADA is incorporated into LipidII in vivo.

Overall, this is an interesting story, with novel finding regards how probes, which are now becoming more widely used in the field, are being incorporated into the PG polymer.

Essential revisions:

1) Some aspects of the statistics should be sharpened or added. Figure 2 – An argument for additivity is not most effectively made with a t-test. In this case, seeing that the drugs in combination are used at the same concentration as they are alone, a probabilistic model (e.g. Bliss scoring) should be used instead. For example, amp does not impede labeling alone, so it is actually additive with imi, as the amount labeled together is almost the same as imi alone. If labeling on two drugs independently is reduced to 50% each, for example, then if they act independently (additive), the combination should label at a reduction of 75%. Rescoring this will strengthen the argument of this section. Figure 3 – Because these measurements were taken together, ANOVA and posthoc tests would be more appropriate than t tests on the pairs. Also, in this figure, the statistical analysis seems to be based on both biological and technical replicates. This is misleading, only biological replicates should be used for the statistics. Figure 5 – the bar plots could use some statistical testing

2) Reviewers raised concerns regards the relevance of side-wall labeling. Please address this. An experiment that highlights the relevance/importance of sidewall labelling in mycobacteria would enhance the impact of your manuscript. What is the physiological importance of sidewall labeling to growth or survival?

3) Related to point 2 above, although the imaging findings in Figure 1 do clearly show sidewall labelling, some clarification of the interpretation of this finding is needed. Because the signal is a gradient from the bright pole labelling to the sidewall, it seems possible that the labelled cell wall appears on the sidewall is due to movement from the pole based on growth, rather than in situ labelling. The labelling time is short, possibly arguing against this possibility, but some clarification is required. Optimally, time-lapse experiments could address this issue, but they may be challenging given the need for fixation for probe visualization. Please comment on this issue and potentially design experiments to address this concern.

4) The data in Figure 5 lack clarity. It is not clear how this finding contributes to the paper and how the interpretation of "stabilization" of the probe by B-lactam treatment adds to the model or modifies the interpretation of the prior figures. Please consider this point and integrate these observations more carefully into your manuscript.

5) Antibiotic titrations for labeling experiments would strengthen some observations. Is it possible that in the imipenem treated cells, destabilization of PG, and cell death is affecting probe uptake? Please address the issue of whether probe uptake is affected by cell death.

6) Importantly, all reviewers have raised concerns with the way you have compiled and presented your manuscript. Two main issues that emerged are (I) this seems like an amalgamation of two stories (side wall labeling) and (FDAA probe uptake and related findings) and (II) the manuscript is hard to follow. Please give these comments careful attention when revising your manuscript. It would seem that the major focus of the manuscript is how FDAAs are taken up in mycobacteria. Perhaps you can start your manuscript with this aspect, develop it carefully using appropriate figures and models. Thereafter, you can discuss how the enhanced understanding of FDAAs developed in the manuscript assists with dissecting sidewall labeling. Revise your introduction and discussion (which is currently unnecessarily long) sections appropriately. Perhaps the title needs some reconsideration, to capture the revised focus of the manuscript. Due to the interdisciplinary nature of the work, additional schematics and summary figures would be helpful (in the manner done for Figure 2A). An information box re the chemistry would also help readability and would provide a format to highlight what is learned about these labels. There are some sections that could be moved to supplementary information to help streamline the results (such as Bocillin labeling, vanc staining, and some detail about polar elongation).

---

## [Author Response]

Essential revisions:1) Some aspects of the statistics should be sharpened or added. Figure 2 – An argument for additivity is not most effectively made with a t-test. In this case, seeing that the drugs in combination are used at the same concentration as they are alone, a probabilistic model (e.g. Bliss scoring) should be used instead. For example, amp does not impede labeling alone, so it is actually additive with imi, as the amount labeled together is almost the same as imi alone. If labeling on two drugs independently is reduced to 50% each, for example, then if they act independently (additive), the combination should label at a reduction of 75%. Rescoring this will strengthen the argument of this section.

We have consulted with a colleague in the UMass Amherst statistics department (now in the Acknowledgements section) to improve our analyses.

We first performed titration experiments to gauge the effects of ampicillin, imipenem and Dcycloserine on alkDA incorporation (see also Essential revision #5). Probe incorporation was not impacted by ampicillin at any of the doses. We next assessed the interaction between imipenem and D-cycloserine by the Bliss independence model. The titrations and rescored data are now in Figures 3C and 3D and the accompanying legend and text.

Figure 3 – Because these measurements were taken together, ANOVA and posthoc tests would be more appropriate than t tests on the pairs. Also, in this figure, the statistical analysis seems to be based on both biological and technical replicates. This is misleading, only biological replicates should be used for the statistics.

We have done an ANOVA followed by Dunnett’s test on the biological replicates. Please see revised Figure 4 (formerly Figure 3) and accompanying legend.

Figure 5 – the bar plots could use some statistical testing

We have done an ANOVA on the log_10_-transformed data from the biological replicates for Figure 4—figure supplement 2 (formerly Figure 5A). The rest of the data from this figure has been removed (see also Essential revision #4).

2) Reviewers raised concerns regards the relevance of side-wall labeling. Please address this. An experiment that highlights the relevance/importance of sidewall labelling in mycobacteria would enhance the impact of your manuscript. What is the physiological importance of sidewall labeling to growth or survival?

In the revised manuscript we now show that lysozyme/mutanolysin treatment increases sidewall dipeptide labeling in *M. smegmatis*. The fluorescence is also more symmetric. We would not expect cell wall damage from muramidases (or other insults) to be limited to the poles and consequently hypothesize that pole-growing organisms must have the ability to monitor and repair the entire cell surface. Please see Figure 6 and accompanying text for more details.

3) Related to point 2 above, although the imaging findings in Figure 1 do clearly show sidewall labelling, some clarification of the interpretation of this finding is needed. Because the signal is a gradient from the bright pole labelling to the sidewall, it seems possible that the labelled cell wall appears on the sidewall is due to movement from the pole based on growth, rather than in situ labelling. The labelling time is short, possibly arguing against this possibility, but some clarification is required. Optimally, time-lapse experiments could address this issue, but they may be challenging given the need for fixation for probe visualization. Please comment on this issue and potentially design experiments to address this concern.

This is what we initially assumed! Movement of the poles may explain some of the brighter sub-polar labeling but we don’t think it can account for labeling that occurs closer to the center of the cell. The original experiments were performed using 15 min pulses of probe incubation, or approximately ~8-10% of *M. smegmatis* generation time. Our pulse chase experiment with RADA labeling (Figure 5—figure supplement 4) illustrates that there is limited movement of the poles in that time frame.

The ideal experiment would be to perform time-lapse imaging with turn-on dipeptide probes to observe incorporation in real time. Unfortunately to our knowledge such reagents have not yet been published. To address the pole movement concern, we instead repeated alkDADA labeling with very short, 2 min pulses (~1% of generation time). We still observe sidewall labeling in addition to the expected polar fluorescence (Figure 5—figure supplement 1).

4) The data in Figure 5 lack clarity. It is not clear how this finding contributes to the paper and how the interpretation of "stabilization" of the probe by B-lactam treatment adds to the model or modifies the interpretation of the prior figures. Please consider this point and integrate these observations more carefully into your manuscript.

The sensitivity of dipeptide labeling to L,D-transpeptidation does not add to our model of sidewall peptidoglycan synthesis in mycobacteria, nor does it change the interpretation of prior figures.

Adding to an already-complicated story, we recently found that imipenem treatment and *ldtE* mutation respectively increase and decrease the abundance of BDL-detectable, lipid-linked precursors. The *ldtE* phenotype is complemented by reintroduction of the gene. While this could potentially be an interesting avenue for future research, we now think that it is premature to present a single model for how L,D-transpeptidases impact alkDADA labeling. At the same time, we do think that this observation is important in the larger context of metabolic labeling interpretation.

To resolve this conundrum, we have pared down the data to show only that alkDADA labeling decreases in *ldt* mutants (now Figure 4—figure supplement 2) but not in the presence of the L,Dtranspeptidase-targeting antibiotic imipenem (Figure 3B,E). We offer several models in the discussion that could explain this finding. Importantly, we also show that the global reduction in signal stems primarily from decreased signal at the poles but leaves sidewall labeling intact (now Figure 4—figure supplement 2B).

5) Antibiotic titrations for labeling experiments would strengthen some observations. Is it possible that in the imipenem treated cells, destabilization of PG, and cell death is affecting probe uptake? Please address the issue of whether probe uptake is affected by cell death.

We have done additional titrations for ampicillin, imipenem and D-cycloserine and alkDA labeling (Figure 3C); titrations for imipenem and HADA, RADA and alkDADA labeling were formerly in Figure 2D, now Figure 3E.

We do not detect cell death in the time frame of the experiment (Figure 3—figure supplement 2). It is more difficult to rule out effects on cell growth or general defects in peptidoglycan metabolism. We have addressed this in Figure 3 by (1) examining probe incorporation in the presence of multiple peptidoglycan-acting antibiotics at different doses and (2) comparing our probes of interest to OalkTMM, which has an independent mechanism of incorporation (Foley et al., 2016).

Two pieces of evidence support a specific effect of imipenem on the monopeptide D-amino acid probes. First, ampicillin and vancomycin do not impact incorporation (Figure 3B,C). This argues against the idea that general peptidoglycan destabilization is responsible for the phenotype. Second, imipenem does not have much effect on alkDADA or OalkTMM incorporation in the time frame of this experiment (Figure 3B,E). We think that it is unlikely that growth inhibition alone would account for the specific effect of imipenem on monopeptide probe incorporation.

6) Importantly, all reviewers have raised concerns with the way you have compiled and presented your manuscript. Two main issues that emerged are (I) this seems like an amalgamation of two stories (side wall labeling) and (FDAA probe uptake and related findings) and (II) the manuscript is hard to follow. Please give these comments careful attention when revising your manuscript. It would seem that the major focus of the manuscript is how FDAAs are taken up in mycobacteria. Perhaps you can start your manuscript with this aspect, develop it carefully using appropriate figures and models. Thereafter, you can discuss how the enhanced understanding of FDAAs developed in the manuscript assists with dissecting sidewall labeling. Revise your introduction and discussion (which is currently unnecessarily long) sections appropriately. Perhaps the title needs some reconsideration, to capture the revised focus of the manuscript. Due to the interdisciplinary nature of the work, additional schematics and summary figures would be helpful (in the manner done for Figure 2A). An information box re the chemistry would also help readability and would provide a format to highlight what is learned about these labels. There are some sections that could be moved to supplementary information to help streamline the results (such as Bocillin labeling, vanc staining, and some detail about polar elongation.

We have revised all sections of the manuscript, as suggested, and included four new schematics (Figures 1, Figure 6A). We appreciate the thoughtful suggestions from the editors and reviewers — the work is now more focused and easier to understand.